# Synaptic plasticity in self-powered artificial striate cortex for binocular orientation selectivity

Yanyun Ren[1,2], Xiaobo Bu[2], Ming Wang[3], Yue Gong[4], Junjie Wang[4], Yuyang Yang[4], Guijun Li [3], Meng Zhang[4], Ye Zhou[2] & Su-Ting Han [4] ✉

Get in-depth understanding of each part of visual pathway yields insights to conquer the challenges that classic computer vision is facing. Here, we first report the bioinspired striate cortex with binocular and orientation selective receptive field based on the crossbar array of self-powered memristors which is solution-processed monolithic all-perovskite system with each cross-point containing one CsFAPbI$_3$ solar cell directly stacking on the CsPbBr$_2$I memristor. The plasticity of self-powered memristor can be modulated by optical stimuli following triplet-STDP rules. Furthermore, plasticity of 3 × 3 flexible crossbar array of self-powered memristors has been successfully modulated based on generalized BCM learning rule for optical-encoded pattern recognition. Finally, we implemented artificial striate cortex with binocularity and orientation selectivity based on two simulated 9 × 9 self-powered memristors networks. The emulation of striate cortex with binocular and orientation selectivity will facilitate the brisk edge and corner detection for machine vision in the future applications.

The history of machine vision spans more than several decades[1]. Nevertheless, the robust and general solutions to the major issues such as motion detection, object recognition, vision-based navigation and activity recognition are still beyond reach of present computer vision system[2]. Biologically visual system is hierarchical organization including retina, optic nerve, lateral geniculate nucleus (LGN) and striate cortex[3–5]. It is evident that the different level of visual system process different types of visual information with receptive fields covering different region of visual field[6]. Get understanding of each part of visual pathway playing in visual perception yields insights of the challenges that classic computer vision is facing[7].

The receptive field is a restricted retinal area where the light shining on it could influence the firing rate of corresponding units[4,8]. The ganglion cells possess concentric receptive field with an "off" center and an "on" border or an "on" center and "off" border. The "on"

and "off" region in receptive field are mutually antagonistic[9]. So that the light spots with circular form and restricted to the "on" area are more effective stimuli for activating retinal ganglion cells[3,10–12]. Its next part, LGN, possesses similar concentric receptive field[13,14]. However, the receptive field of cells in the striate cortex is narrow, long, vertically oriented region which differs strikingly from that in retinal ganglion cells and LGN[15]. Therefore, the vertical slit-shaped spot of light superimposing on the center of receptive field of striate cortical cells often evoke brisk response[16–18]. The key role of receptive field of striate cortex is supporting the edge and corner detection so the motion detection is usually processed here[19].

Along the visual pathway, the receptive field of different level of visual system shows convergence process[20]. The receptive field of one ganglion can be viewed as the collective of the many photosensory cells which synapses with it. In turn, the group of ganglion cells form

[1]Institute for Microscale Optoelectronics, Shenzhen University, Shenzhen 518060, PR China. [2]Institute for Advanced Study, Shenzhen University, Shenzhen 518060, PR China. [3]Key Laboratory of Optoelectronic Devices and Systems of Ministry of Education and Guangdong Province, College of Physics and Optoelectronic Engineering, Shenzhen University, Shenzhen 518060, PR China. [4]College of Electronics and Information Engineering, Shenzhen University, Shenzhen 518060, PR China. ✉e-mail: sutinghan@szu.edu.cn

the receptive field of striate cortical cells (Fig. 1a). Therefore, the receptive field of one striate cortical neuron can be viewed as the collective of the many retinal photosensory cells (rods and cones) which indirectly synapse with it[21]. The modulation of synaptic connections between the photosensory cells and striate cortical cells is crucial to develop the narrow, slit-shaped receptive field, ensuring the orientation selective in striate cortex[22,23].

Compared with Purely Hebbian, that is the modification of synapse based on the multiplication of the pre- and post-stimuli and the synaptic weight stabilized by controlling cortical responses below the maximum, the rate-based Bienenstock-Cooper-Munro (BCM) learning rule is more biorealistic for the synaptic modification and neuronal response selectivity in the experience-dependent modification that observed in striate cortex[24]. It describes that the sign of synapse weight modification is determined by whether the postsynaptic response exceeds a threshold. The postsynaptic firing rate higher than a sliding threshold induces the strengthening of synapse while the postsynaptic responses below the threshold weaken the synapse[25,26]. The sliding threshold is dependent on the average activity of the postsynaptic neuron, ensuring a history- or experience-dependent characteristic, which is the figure of merit of BCM[27–29]. In order to realize BCM, a triplet-STDP which introduces a third presynaptic or postsynaptic spike to pair-STDP has been employed to reproduce frequency effect of the pair protocol[30,31]. The frequency effect stems from the pair spikes-induced paired term and the previous spike of triplet-STDP induced triplet term[32]. Furthermore, the triplet-STDP can be employed to realize rate-based BCM learning rule through an All-to-All framework[33].

Except for a strong preference for a particular orientation, the visual response of striate cortical neuron is also binocular[34]. In the biological visual system, signals from the left and the right eyes first converge in the striate cortex, V1. Neurons in adult striate cortex are binocular with a strong preference for contours of a particular orientation. In biology visual system, the newly born interocular neurons have different orientation preference which means the binocular response is inconsistent for the same field of view (FoV). A matching process between two eyes is required to form normal binocular perception for depth and stereopsis[35]. Binocular neurons in the striate cortex must match their orientation tuning through the two eyes in order for the animal to perceive coherently.

Currently, the vision sensors are emerging to mimic receptive field of ganglion cell in retina for simultaneously sensing and processing. The emulation of experience-dependent modifications of synaptic strength to form binocular, orientation selective receptive field that observed in the striate cortex lags considerably behind that of the retina[36–38]. In this work, we first report the bioinspired striate cortex with binocularity and orientation selectivity based on the crossbar array of self-powered memristors where each cross-point contains one CsFAPbI$_3$ perovskite solar cell and one CsPbBr$_2$I perovskite memristor. The second-order CsPbBr$_2$I memristor with mobile halogenic vacancy similar to Ca$^{2+}$ dynamics in the striate cortical synapse, which allows the emulation of rate-based plasticity. While the CsFAPbI$_3$ perovskite solar cell can be viewed as photosensory retinal cells to synapse with striate cortical cell for converting external optical signals into electrical signals (For biological visual system, the cortical cell synapses with LGN, LGN synapses with ganglion cells, and ganglion cell synapses with retinal sensory cells. In our hardware implementation, we directly synapse artificial retinal photosensory cells with cortical cell for simplicity). Furthermore, plasticity of 3 × 3 crossbar array of self-powered memristor has been successfully modulated based on rate-based BCM learning rules for pattern recognition by light illumination. Finally, we realized artificial striate cortex with binocularity and

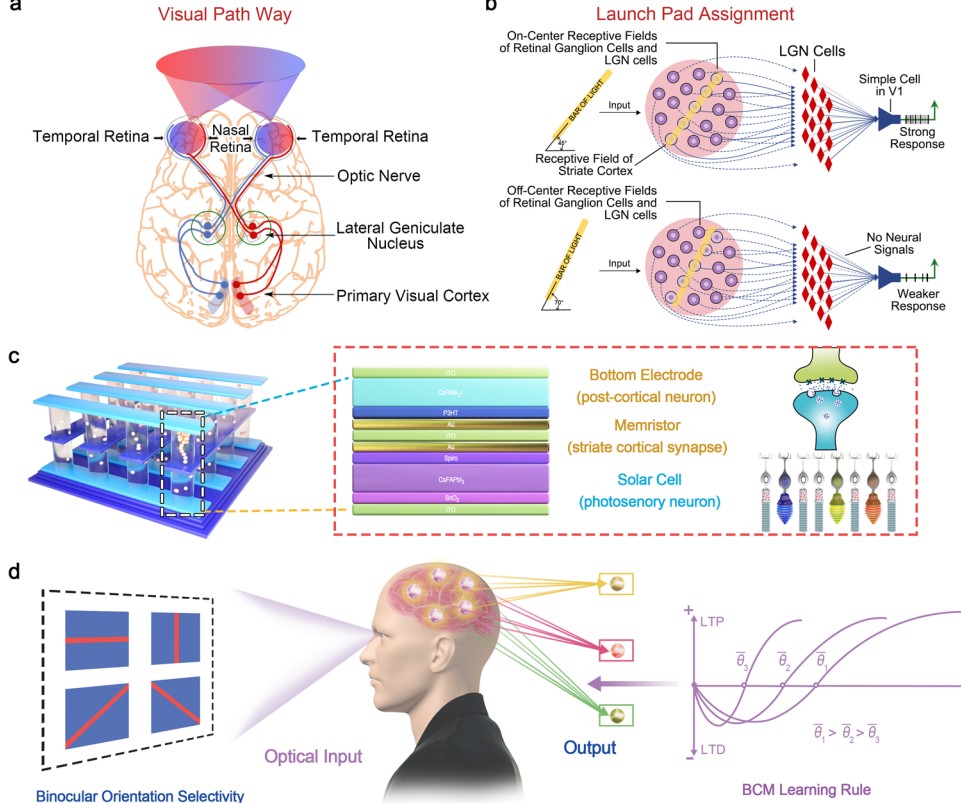

**Fig. 1 | Artificial striate cortex based on self-powered memristor. a** The hierarchy of human visual system for light perception and processing. **b** The convergence process of receptive field in different level of visual system. **c** The concept of artificial striate cortex based on self-powered perovskite memristor. **d** The development of narrow, slit-shaped receptive field of striate cortex with binocular orientation selectivity based on BCM learning rule.

orientation selectivity based on two $9 \times 9$ self-powered memristor networks, following the generalized BCM learning rule. By varying the type of input for (1) normal binocular contour vision, (2) monocular deprivation, (3) binocular deprivation, we highly reproduced the experience-dependent modifications that have been observed experimentally in kitten striate cortex. To the best of our knowledge, this is the first time to realize the hardware implemented striate cortex with binocular and orientation selectivity. The bio-inspired striate cortex is highly compatible with high-density and low power consumption machine vision owing to its crossbar paradigm and homotypic materials system.

## Results

### The concept of artificial striate cortex

As mentioned above, the retina (photoreceptor layer), bipolar cell layer, horizontal cell layer and ganglion cell layer, optic nerve, lateral geniculate nucleus (LGN) and striate cortex are organized in a hierarchical way to realize visual perception (Fig. 1a). In the retina, the photoreceptors (rod opsins and cone opsins) detect the outside stimuli and directly synapse with bipolar cells, which in turn synapse with ganglion cells and conduct action potentials to the LGN. The LGN consisting of six layers in humans is a sensory relay nucleus in the thalamus of the brain[39]. The striate cortex, which is also called primary striate cortex (area V1) receives and organizes information from LGN and further sends the signal downstream to higher visual area[19]. The striate cortex is the first cortex area to receive visual information in which the cells are more effectively stimulated by bars or edges with narrow ranges of orientations. The complete map of receptive field of striate cortex cells is shown in Fig. 1b with dark purple pad areas giving excitation and surrounding light purple circle areas giving inhibitory effects. When the vertically slit-shaped spot of light covering the receptive field located in central area of eye hemisphere, the greatest "on" response accompanies the strongest inhibitory responses gives the peak response of striate cortex with highest firing rate[3,4]. The key role of receptive field of striate cortex is supporting the edge and corner detection so the motion detection is usually processed here[19].

Here we emulate the striate cortex with binocular and orientation selective receptive field by integrating self-powered memristor into a crossbar array where the plasticity of the individual device is modulated by light pulses separately (Fig. 1c). Each cross-point contains one two-terminal $CsFAPbI_3$ perovskite solar cell directly stacking on the two-terminal $CsPbBr_2I$ perovskite memristor. The solar cell functions as pre-synaptic photosensory neuron in retina, converting the external optical signals to electrical signals. The memristor functions as striate cortical synapse which can be driven by the solar cell. The bottom electrode (BE) of self-powered memristor can be regarded as postcortical neuron and the change of conductance emulates potentiation or depression of synaptic weight for the development of binocular and orientation selective receptive field. According to the BCM rate-based theory, the plasticity of excitatory cortical synapses varies as the product of input activity and a function of the summed postsynaptic response ($\phi$). $\phi$ exhibits a negative value for postsynaptic response less than modification threshold ($\theta$) while $\phi$ has a positive value for postsynaptic response greater than $\theta$. The novel feature of the BCM theory is that the value of $\theta$ is not fixed which slides as a nonlinear function of the average postsynaptic responses, allowing the precise specification of biological situation[40]. Figure 1d shows the schematic illustration of development of binocular, orientation preference receptive field in the human visual system driven by visual experience based on the BCM rate-based theory with typical feature of sliding threshold.

### Synaptic emulation of $CsPbBr_2I$ memristor

First, the pure perovskite memristor with structure of Au/P3HT(poly(3-hexylthiophene))/$CsPbBr_2I$/ITO were fabricated to verify its second order effect for emulating rate-based plasticity. Figure 2a shows the

schematic illustration and the cross-sectional scanning electron microscopy (SEM) image of the perovskite memristor. The ultrathin P3HT layer functions as the passivation layer to decrease the defects of perovskite and reservoir layer to retain halogenic ions. The thickness of perovskite film in memristor is ~100 nm. The X-ray photoelectron spectroscopy (XPS), X-ray diffraction (XRD), atomic force microscopy (AFM) characteristics of perovskite film are in Supplementary Fig. 1. The P3HT film is too thin to be directly characterized by scanning electron microscopy (SEM) and its existence can be verified by the infrared transmittance spectra[41]. For electrical measurements of memristor, the external voltage approaches on the Au TE while the ITO BE keeps ground, the input voltage pulse, device conductance and response current are regarded as the presynaptic spike, synaptic strength and postsynaptic current. The direct current (DC) measurement of memristor shows the pinched hysteresis phenomenon, as shown in Fig. 2b. Different from the bistable resistive switching phenomenon, the response current of our device gradually increases during application of five consecutive positive sweeps and decreases followed five consecutive negative sweeps, indicating tunable conductance characteristics which is analogue to the modulation of biological synapses.

In Supplementary Fig. 1d, we demonstrate the excitatory postsynaptic current (EPSC) behavior through memristor under a single presynaptic spike, the EPSC increase abruptly at the arrival of presynaptic pulse which takes about 200 ms to relax back to the initial state. The abrupt EPSC increase is generally attributed to the voltage-triggered halogenic vacancy (e.g. bromine vacancy ($V_{Br}$) and iodine vacancy ($V_I$)) migration and the hysteresis phenomenon may be originated from the spontaneous diffusion of halogenic vacancy after removal of external bias[42–44]. Hence, our memristor displays second-order phenomena with first-order variable of halogenic vacancy distribution into a steady state in long term dynamics and the second-order variable of spontaneously redistribution of halogenic vacancy. Furthermore, the paired-pulse facilitation (PPF) is an important short-term phenomenon in neuroscience which represents a facilitation effect of EPSC when the pre-synaptic membrane is stimulated by two consecutive spikes. The second spike will trigger a reinforced EPSC on the basis of first spike where the amplitude enhancement is determined by the interval time between two spikes. Figure 2c shows the PPF experimental results and the PPF value decreases with the increase of interval time, the experiment data can be defined as:

$$PPF = c_1 e^{-\frac{t}{\tau_1}} + c_2 e^{-\frac{t}{\tau_2}} \qquad (1)$$

where the $c_1/c_2$ and $\tau_1/\tau_2$ are the initial magnitudes and characteristic relaxation times of the rapid and slow phases, respectively[45]. By fitting the PPF curve, the $\tau_1$ of short-term potentiation (STP) constant and $\tau_2$ of long-term potentiation (LTP) constant were estimated as 0.28 ms and 10.86 ms, respectively. The $\tau_2$ is one order larger than $\tau_1$ which is comparable to the biological synapse (~ms order)[46,47]. The fast relaxation process is originated from the low activation energy of halogenic vacancy migration[43].

The programmable conductance levels of memristor are essential toward mimicking long-term synaptic plasticity such as LTP and long-term depression (LTD)[48,49]. Figure 2d shows that the memristor conductance can be gradually increased (decreased) by consecutive 100 positive 0.5 V/5 ms (negative −0.5 V/5 ms) pulses, which is corresponding to the synaptic LTP and LTD behaviors. To quantitative assessing the plasticity of our memristor, the conductance window ($G_{max} - G_{min}$) and the non-linearity (NL) of LTP/LTD are calculated as 3.00/3.98 (Note Non-Linearity in Supporting Information). We also characterized the current responses by separately varying the pulse interval and pulse width as shown in Fig. 2e, f. It is clear that a stronger LTP behavior can be obtained under pulse train stimuli with shorter pulse interval and larger pulse width. When the pulse interval

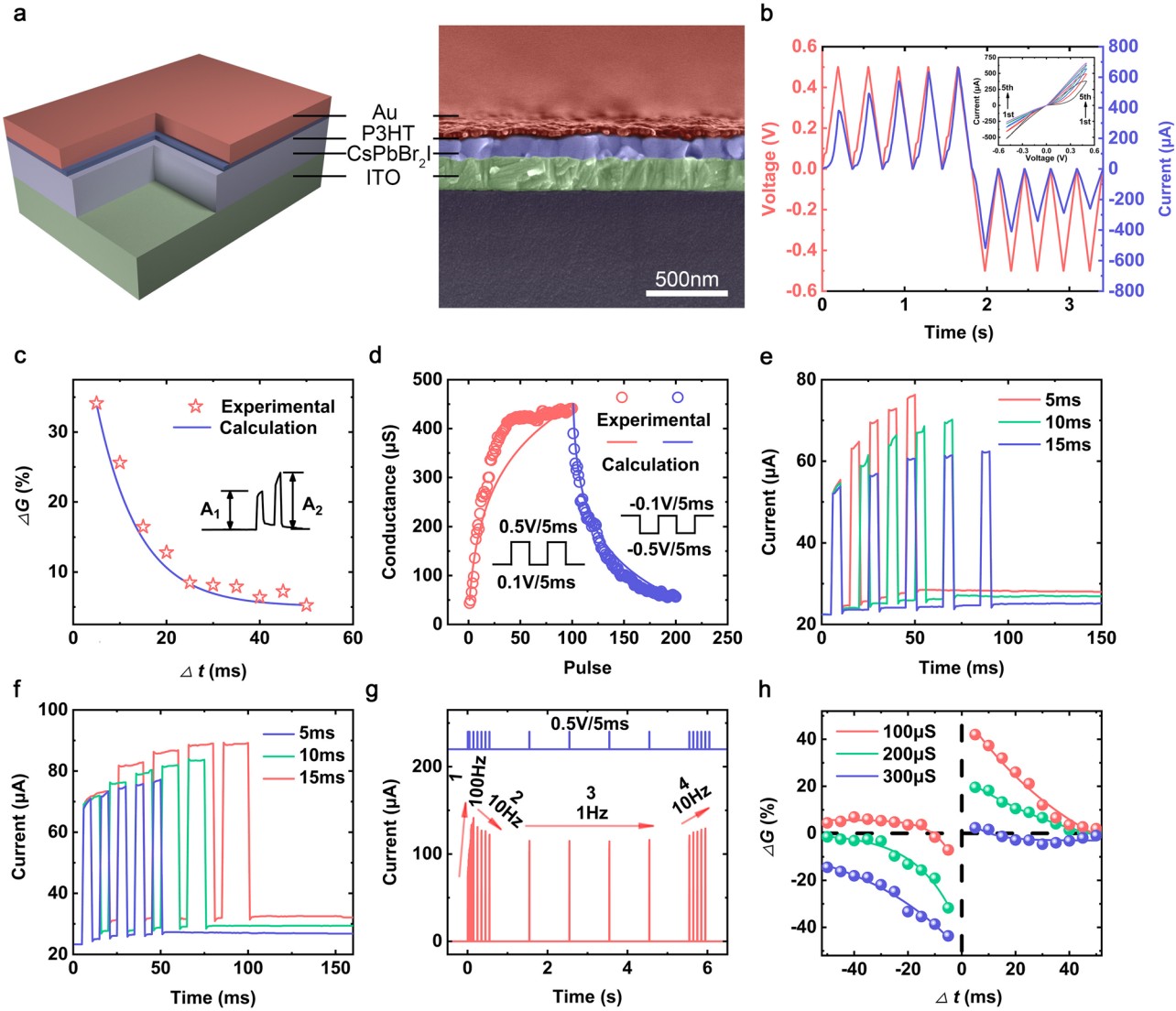

**Fig. 2 | Emulation of artificial synapses. a** The schematic and the cross-sectional SEM image of Au/P3HT/CsPbBr$_2$I/ITO memristor. **b** The current with respect to DC voltage sweeps, the device conductance gradually increase (decrease) under positive (negative) sweeps of 0.5 V (−0.5 V). **c** The PPF ratio as a function of pre-synaptic pulse interval which can be expressed as $(A_2 - A_1)/A_1 \times 100\%$. The pulse width and amplitude were set to 5 ms and 0.5 V. **d** The modulation of device conductance as a function of 100 consecutive potentiating and 100 depressing pulses. **e** Emulation of EPSC versus pulse interval at 5 ms, 10 ms and 15 ms. **f** Emulation of EPSC versus pulse width at 5 ms, 10 ms and 15 ms. **g** The device response to consecutive presynaptic pulse sequences at the frequencies of 100 Hz, 10 Hz, 1 Hz and 10 Hz. **h** The STDP learning rule simulated at three different initial states ($G_0 = 100$ μs, 200 μs, 300 μs) as a function of presynaptic and postsynaptic pulses interval.

decreases form 15 ms to 5 ms, the peak current increases from 62.1 μA to 76.3 μA.

Synaptic plasticity not only depends on the competition between excitatory and spontaneous decay, but also relates to the previous activity[50,51]. This history-dependent plasticity can be realized based on the perovskite memristor, as shown in Fig. 2g. The four pulse trains with fixed pulse amplitude of 0.5 V, fixed width of 5 ms and different frequencies were applied to memristor successively. It is clearly that, the device conductance rapidly increases after the first pulse train with frequency of 100 Hz, indicating that the excitatory of device overcomes its spontaneous decay. While the conductance decreases under second pulse train with low spike frequency of 10 Hz, implying that the spontaneous decay is dominated. During the third pulse train with the frequency of 1 Hz, the device conductance remains essentially unchanged because the relaxation and excitatory process of device achieve balanced state at low spike frequency. Interestingly, the fourth pulse train with spike frequency of 10 Hz triggers the increase of conductance, displaying opposite trend of conductance modulation

compared with the application of second pulse train even though the same spike frequency were employed. It can be explained that after relaxation of device during third pulse train, the excitatory of device re-dominate the conductance modulation under the fourth pulse train with relatively higher spike frequency. In brief, the different modulation of conductance change can be obtained at a given stimulation condition (e.g. 10 Hz) based on different previous activities (e.g. 100 Hz, 1 Hz), indicating that our memristor can simulate history-dependent synaptic plasticity successfully.

Along with the single-terminal-triggered synaptic plasticity (e.g. PPF, LTP and LTD), the pair-STDP which describes a plasticity modulation by adjusting the temporal intervals between presynaptic and postsynaptic pulses[45,52]. In order to verify the feasibility of the pair-STDP emulation for our memristor, a pair of rectangular voltage pulse with amplitude of 0.5 V, width of 5 ms and variable pulse interval ($\Delta t = t_{post} - t_{pre}$) were applied to the pre-synaptic neuron and post-synaptic neuron. Based on the pair-STDP learning rule, if the pre-synaptic pulse arrived before postsynaptic pulse, ($\Delta t > 0$), the synaptic

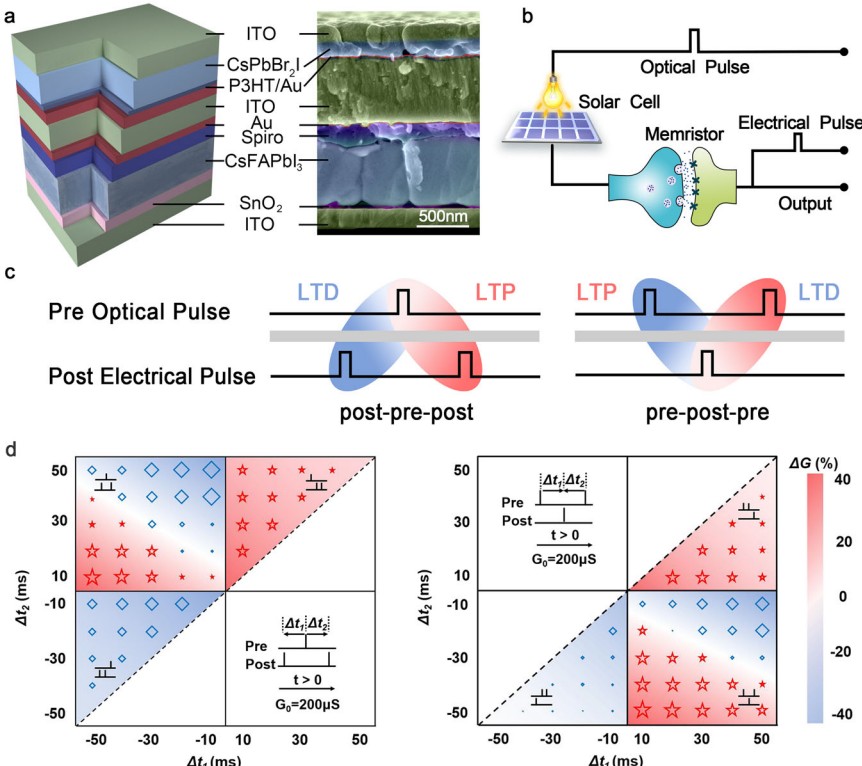

**Fig. 3 | The triplet-STDP implement of self-power memristor under light stimuli. a** The schematic and the cross-sectional SEM image of self-powered memristor. **b** Circuit diagram of self-powered memristor for realizing triplet-STDP. **c** The schematic of the typical triplet-STDP, including 'post-pre-post' and 'pre-post-pre' pulse sequences. **d** The triplet-STDP simulation based on self-powered memristor.

The memristor was preset to a middle conductance 200 μs and the experimental results show that the potentiation or depression is determined by the pulses sequences and its timing intervals. The size of red star (potentiation) and blue rhombus (depression) with corresponding background color are employed to express the magnitude of device conductance change.

weight increases to induce LTP. On the contrary, if the order of pairing pulse reverses, ($\Delta t < 0$), the synaptic weight decreases to obtain LTD. In addition, the amplitude of synaptic weight change ($\Delta G$) shows a negative exponent relationship with pulse interval, where the pair-pulse with a smaller pulse interval usually results in larger LTP or LTD. Inspired by the history-dependent plasticity as shown in Fig. 2g, we emulated the pair-STDP plasticity in three initial states with $G_0$ of 100 μs, 200 μs and 300 μs, as shown in Fig. 2h. The sign of $\Delta G$ changes from positive to negative with the increase of $\Delta t$ in high $G_0$ state, and vice versa. This phenomenon can be explained as when pair-pulse with a large $\Delta t$ is employed to activate the device, the increase of conductance fails to overcome the decay effect. Hence, the conductance decreases even in a conventional LTP region. This successful simulation of history-dependent pair-STDP learning rule is the foundation of realizing BCM learning rules in our self-powered memristor, more details are shown in Supplementary Fig. 17.

## Triple-STDP in self-powered memristor

In order to realize BCM learning rule, a triplet-STDP which introduces a third presynaptic or postsynaptic spike to pair-STDP has been employed to reproduce frequency effect of the pair protocol[53,54]. In this part, light modulation of plasticity of self-powered memristor with structure of ITO/CsPbBr$_2$I/P3HT/Au/ITO/Au/Spiro/CsFAPbI$_3$/SnO$_2$/ITO were realized based on triple-STDP (Fig. 3a). By delicately compound engineering and structural design, we first realize solution-processed monolithic all-perovskite system to implement the self-powered memristor for realizing hardware-based striate cortex. The microscopic images were obtained by performing the cross-sectional scanning electron microscopy (SEM) on device where the continuous and smooth surface of each layer can be observed, verifying that the structure contains one CsFAPbI$_3$ perovskite solar cell directly stacking

on the CsPbBr$_2$I perovskite memristor and our design of solution-processed monolithic all-perovskite system allows the facile integration of memristor and solar cell without damaging of each layer. The solar cell functions as pre-synaptic photosensory neuron in retina, generating a voltage spike with the open-circuit voltage ($V_{oc}$) of ~1 V under light irradiance of 100 mW/cm$^2$ (Supplementary Figs. 7–9). The photovoltaic potential induced by photoelectric conversion activates the memristor which is viewed as striate cortical synapse. The BE of self-powered memristor is regarded as post-cortical neuron and the change of conductance emulates potentiation or depression of synaptic weight obeying triplet-STDP learning rule (Fig. 3b).

For the long-term triplet-STDP, the spike train can be considered as the integration of two spike pairs which induce the combination of LTP and LTD processes and the synaptic modification (Fig. 3c)[31,55]. For example, in 'post-pre-post' sequence, the first 'post-pre' pair ($\Delta t_1 = t_{post}^1 - t_{pre} < 0$) of 'post-pre-post' pulse induces a LTD, following by a LTP caused by the second 'pre-post' pair ($\Delta t_2 = t_{post}^2 - t_{pre} > 0$). Similarly, the 'pre-post-pre' configuration can also be divided as a 'pre-post' pair (LTP, $\Delta t_1 = t_{post} - t_{pre}^1 > 0$) and a 'post-pre' pair (LTD, $\Delta t_2 = t_{post} - t_{pre}^2 < 0$). Moreover, the triplet term made by the previous spike to the paired spikes is required to be taken into the consideration. Figure 3d summarize the data of the first-spike-dominating triplet-STDP (the first spike suppresses the efficacies of subsequent spikes) with varying $\Delta t_1$ and $\Delta t_2$ using a colored background to show $\Delta G$[30,56]. The results in quadrant **II** and **IV** regions are the typical triplet-STDP. In the quadrant **II** region, the synapse depression and potentiation are realized simultaneously in the 'post-pre-post' pulses by adjusting the parameters of $\Delta t_1$ and $\Delta t_2$. As shown in the up-right part, the $|\Delta t_1|$ is smaller than $|\Delta t_2|$, the LTD dominates the whole process which results in the weight depression. With the increase of $|\Delta t_1|$, the difference of $|\Delta t_1|$ and $|\Delta t_2|$ goes down, the equilibrium state received at

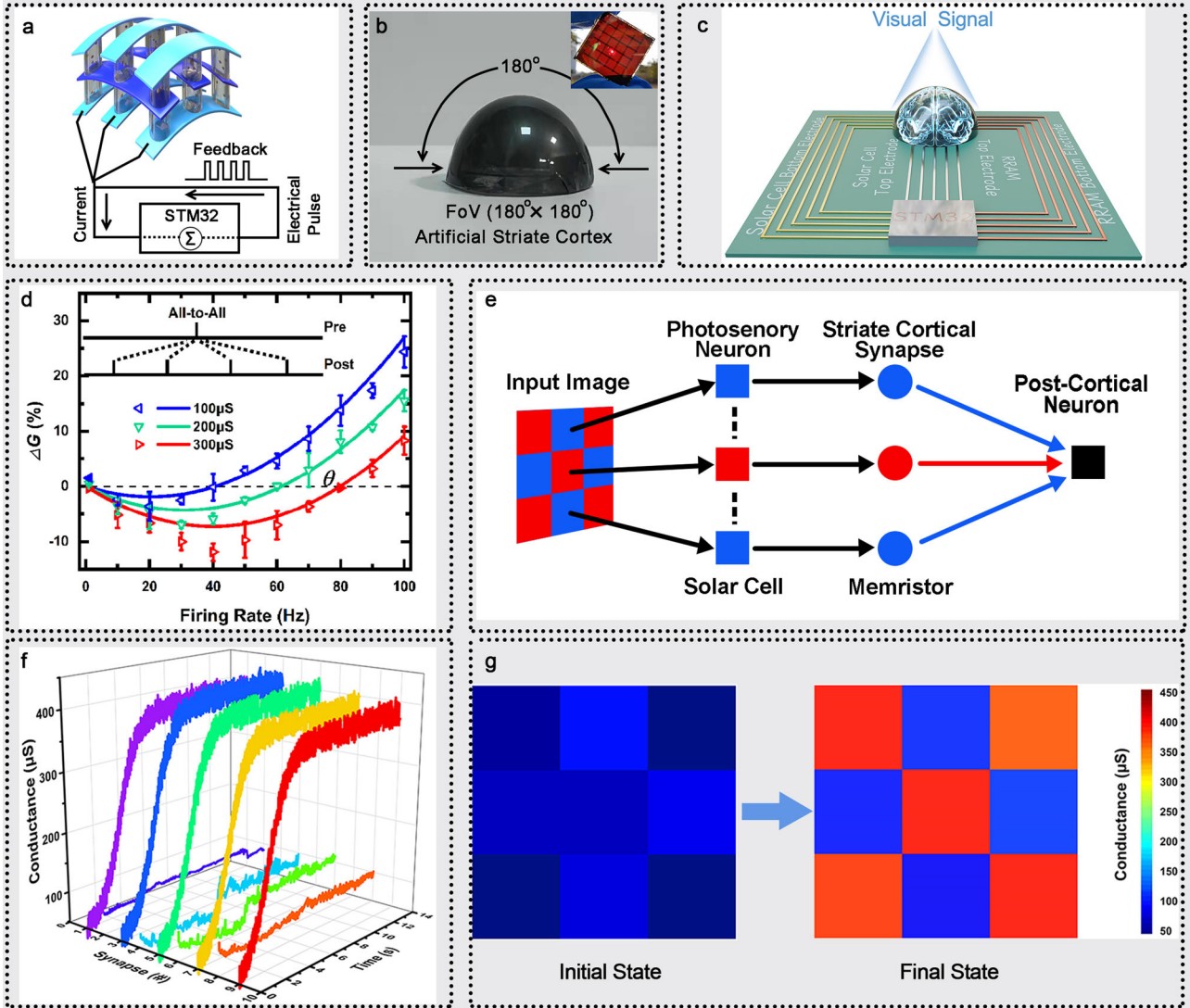

**Fig. 4 | BCM learning rule for pattern learning based on device-level artificial striate cortex. a** The schematic illustration of artificial striate cortex and the postsynaptic electric circuit design. **b** The optical images of hemispherical-shaped artificial striate cortex and the flexible array glowing as sunlight passes through it (insert). And the dimension of the array is 3 cm × 3 cm with the electrode width of 1000 μm, where the large array size is limited by the size of sub-solar cell for

receiving sufficient visual information from the environment. **c** The schematic of circuit block. **d** The measured (triangle) and fitting (line) results of BCM learning rule as a function of postsynaptic firing frequency based on the artificial striate cortex. **e** The schematic of artificial striate cortex for pattern learning. **f** The evolution of synaptic weights of the 9 visual cortical synapses during pattern learning task. **g** The initial and final states of synaptic weights in the pattern learning task.

the state of $|\Delta t_1| \approx |\Delta t_2|$. When the $|\Delta t_1|$ continuously increases to be larger than $|\Delta t_2|$, the LTP plays a dominant role reversely and results in weight potentiation (the bottom-left part). In the quadrant **IV** region of 'pre-post-pre' sequence, the weight depression and potentiation take place in the up-right ($|\Delta t_1| > |\Delta t_2|$) and bottom-left ($|\Delta t_1| < |\Delta t_2|$) parts. It is worth pointing out that, the conversion of depression and potentiation does not appear in the $|\Delta t_1| \approx |\Delta t_2|$ region owing to the decay effect.

The asymmetry characteristic of triplet-STDP is beneficial to the following realization of BCM-rate learning rule compared to pair-STDP. Beside the typical triplet-STDP, four other triplet-STDP (e. g. 'pre-pre-post', 'pre-post-post', 'post-post-pre', 'post-pre-pre') are also performed in quadrant **I** and **III** regions, where the presynaptic pulses are always received before (after) postsynaptic pulses. Actually, the quadrant **I** region can be simplified to LTP behavior and potentiation effect is observed while the quadrant **III** region can be simplified to LTD behavior and pure depression. The fitting results of triplet-STDP are shown in Supplementary Table 6, which can support us to further implement the BCM learning rule quantitatively, e.g. the weight change value in BCM learning rule is

determined by the triplet potentiation/depression terms of triplet-STDP.

## Pattern learning in self-powered artificial striate cortex

Next, to demonstrate the functionality of self-powered memristor for developing the receptive field of striate cortex based on BCM rate-based learning rule, the flexible 5 × 5 self-powered memristor array were fabricated. As illustrated in Fig. 4a, nine solar cells in the crossbar array function as pre-synaptic photosensory neuron in retina to emit voltage spike trains corresponding to the input visual pattern. The nine memristors are viewed as striate cortical synapses and the bottom electrodes of nine self-powered memristors were connected together as one post-cortical neuron. The Fig. 4b shows the optical image of hemispherical shaped artificial striate cortex with field-of-view (FoV) of 180° along both the x and y direction and the inset shows the optical image of the flexible array glowing as sunlight passes through it, indicating the high flexibility and transparency of the array. The crossbar array of self-powered memristors were further connected into the processing unit STM32 which has been integrated in printed

circuit board (PCB), as shown in Fig. 4c and Supplementary Fig. 18. All electrodes including bottom electrodes of solar cells, top electrodes of solar cells which are also the top electrodes of memristors and bottom electrodes of memristors are connected to controller, therefore all the synaptic states can be controlled and monitored in real time. During the learning process, the input voltage pulse is generated by solar cell when optical pulse arrives with a rate of $\rho_x$ (Poisson distribution). Each input pulse is applied to TE of memristor and generates a post neuron current which collected by STM32, where the total current is $I = \sum_{t<200ms} w_0 G$ where $w_0 = 0.5$ V. Then, the STM32 feedback electrical pulse to BE of memristor with firing rate according to $\rho_y = g \times I$, where $g = 50$ Hz/mA. In this framework, simplified the constants, the numerical value of postsynaptic firing rate can be redefined as $\rho_y = G\rho_x$ with a Poisson distribution, too. Hence, the synaptic weight change induced by the BCM learning rule with the presynaptic input rate $\rho_x$ and post-synaptic firing rate $\rho_y$[57–59].

The triplet-STDP methods can be employed to attain the generalized BCM learning rule in our self-powered memristor array for high-order spatiotemporal recognition. The BCM learning exhibits two features. Firstly, if the synaptic input with firing rate $\rho_x$ drives post-synaptic firing rate $\rho_y = G\rho_x$ to a high level, the potentiation effect occurs while if the synaptic input drives $\rho_y$ to a low level, the depression effect can be obtained which is called enhanced depression effect. Secondly, a modification threshold $\theta$ is defined as the crosspoint between potentiation and depression. It shows sliding characteristic that a potentiation direction moves when average postsynaptic firing rate $\langle\rho_y\rangle$ is low, while a depression direction moves when $\langle\rho_y\rangle$ is high. Based on a single linear neuron assumption, the synapse weight modification based on BCM learning rules can be simplified as:

$$dG/dt = \phi(\rho_y, \theta)\rho_x \tag{2}$$

where $\phi()$ is a scalar function of $\rho_y$ and $\theta$. Thus, the BCM learning rule can be simplified as a function of postsynaptic pulse firing rate, slide threshold and presynaptic input pulse rate[40,60].

Indeed, the change of synaptic weight base on BCM learning can be calculated based on All-to-All framework (the synaptic change is the integration of changes made by all possible pre- and post-synaptic pairs) based on our self-powered memristors array. In this framework, the weight change is determined by the interaction of every single spike with all other spikes depend on triplet-STDP:

$$dG/dt = \phi(\rho_y, \theta)\rho_x = (-A_2^-\tau_-\rho_y - A_3^-\tau_-\tau_x\rho_x\rho_y + A_2^+\tau_+\rho_y + A_3^+\tau_+\tau_y\rho_y^2)\rho_x \tag{3}$$

where the $A_2^+, A_2^-, A_3^+, A_3^-$ are the relevant amplitudes, and the $\tau_+, \tau_-, \tau_x, \tau_y$ are the corresponding time constants of potentiation and depression terms in pair-STDP and triplet-STDP. This fitting function can satisfy the BCM learning through parameters deformation as follows. On the one hand, the $A_3^- = 0$ is used to emulate the first feature of BCM based on a minimal triplet rule, namely, $(\rho_y < \theta, \theta) < 0$, $(\rho_y > \theta, \theta) > 0$ and $(0, \theta) = 0$. On the other hand, the second feature is matched by redefining $A_2^- \rightarrow A_2^-\langle\rho_y^2\rangle/\rho_0^2$, $A_2^+ \rightarrow A_2^+\langle\rho_y^2\rangle/\rho_0^2$ where $\rho_0$ is a constant. This yields a frequency dependent threshold $\theta = \langle\rho_y^2\rangle(A_2^-\tau_- - A_2^+\tau_+)/(\rho_0^2 A_3^+\tau_+\tau_y)$ which is proportional to the expectation over the second power of postsynaptic firing rate[37,55]. For our self-powered memristors array, threshold sliding effect can be realized by tuning synapse history state $G_0$ as shown in Fig. 4d. The memristor display the depression effect at low postsynaptic firing rate and potentiation effect at higher firing rate. The threshold slides from 40 Hz, 60 Hz to 80 Hz with the increase of synapse history states from 100 μs, 200 μs to 300 μs. It should be pointed out that, this BCM implementation is based on the assumption that the responsiveness of neuron connectivity is active which means pre-synapse and post-synapse need keep fires to active the retina-genicular-cortical pathway.

In order to demonstrate that the BCM learning rule allows the development of receptive field of artificial striate cortex, the simple visual pattern learning was first implemented based on the crossbar array of self-powered memristors. The visual pattern was encoded into optical pulses which acts on the solar cell (artificial photosensory neurons), the solar cells then convert the optical stimuli into electrical pulses to activate the memristor. The current generated in each memristor was collected by the post-cortical neuron (Fig. 4e). The red pixels of the input pattern 'X' possess high optical input rate of 30 Hz while the blue pixels of input pattern possess low optical input rate of 14 Hz. The post-cortical neuron responds to the input singles following a linear function with a firing rate of $\rho_y = G\rho_x = \sum_{i=1}^{9} G_i\rho_x^i$, where $G = [G_1, G_2 \cdots G_i]$ and $\rho_x = [\rho_x^1, \rho_x^2 \cdots \rho_x^i]$ ($i$ is the index of the synapse) are the synapse weight and input rate, respectively. The synaptic weights modulation and pattern learn result are shown in Fig. 4f, g. All the synapses are set to lower stochastic weights as the initial state. enduring the learning process, the pattern pixels are stimulated to high weights (synapse 1, 3, 5, 7, 9) owing to the high input rate, while the background pixels (synapse 2, 4, 6, 8) possess low weights. In the end, a well-defined 'X' pattern has been mapped to the artificial striate cortex.

### Artificial striate cortex with binocularity and orientation selectivity

To show the potential of the self-powered memristor in the construction of large-scale crossbar array for realizing artificial striate cortex with binocularity and orientation selectivity, we performed a simulation based on experimental data. The BCM learning rule is attractive because it is suitable for explaining the development of spatiotemporal receptive field properties encountered in striate cortex (V1)[61,62]. The orientation selectivity, which means the neurons may exhibit high firing rate spike corresponding to the input bar with a particular orientation while low firing rate spike is obtained corresponding to the input with other orientations[16,63]. Nowadays, most of the experimental and simulation works have been focused on electrical monocular simulation[64,65]. The critical light sensitivity characteristic as well as the binocular rearing correlation (e.g. binocular competition effect) of biology visual system have not been demonstrated yet.

Our self-powered memristors array is an ideal candidate to simulate orientation selectivity and binocularity in striate cortex owing to (1) the exact implementation of BCM learning rule and (2) the great functional congruent with biological visual system (discussed in last section)[66]. As shown in Fig. 5a, the 2-layer spiking neural network (SNN) including two 9 × 9 resolution input layer and a single cell output layer was design to represents the two eyes and one striate cortical neuron, respectively. Four different orientation bars in the tilt angle of 0°, 45°, 90°, 135° were used as input patterns, the grids on the orientation bar possess high optical input rate of 30 Hz while other grids possess low optical input rate of 14 Hz. The cortical cell responds to the input singles following a linear function with a firing rate of $\rho_y = G_l\rho_x^l + G_r\rho_x^r$, where $G_l/G_r$ and $\rho_x^l/\rho_x^r$ are synapse weight and input rate of left and right eyes, respectively. Before the simulation, all the synapse is given lower stochastic weight. During each simulation epoch, the four orientation patterns are stochastically input to the right and left eyes with equal probability, and the postsynaptic neuron synchronously responds based on the real-time presynaptic input with the firing rate of $\rho_y$. Then, the synaptic weight $G$ and the threshold $\theta$ are modified following BCM learning rule in real-time. Finally, the orientation bar which can drives the highest postsynaptic firing rate will selected by the network, serving as the winner orientation and inhibiting other orientations shake its winner state. The detailed simulation flow chart is shown in Supplementary Fig. 20.

Based on the designed SNN network, the orientation selectivity simulation is executed in the conditions of (1) normal binocular

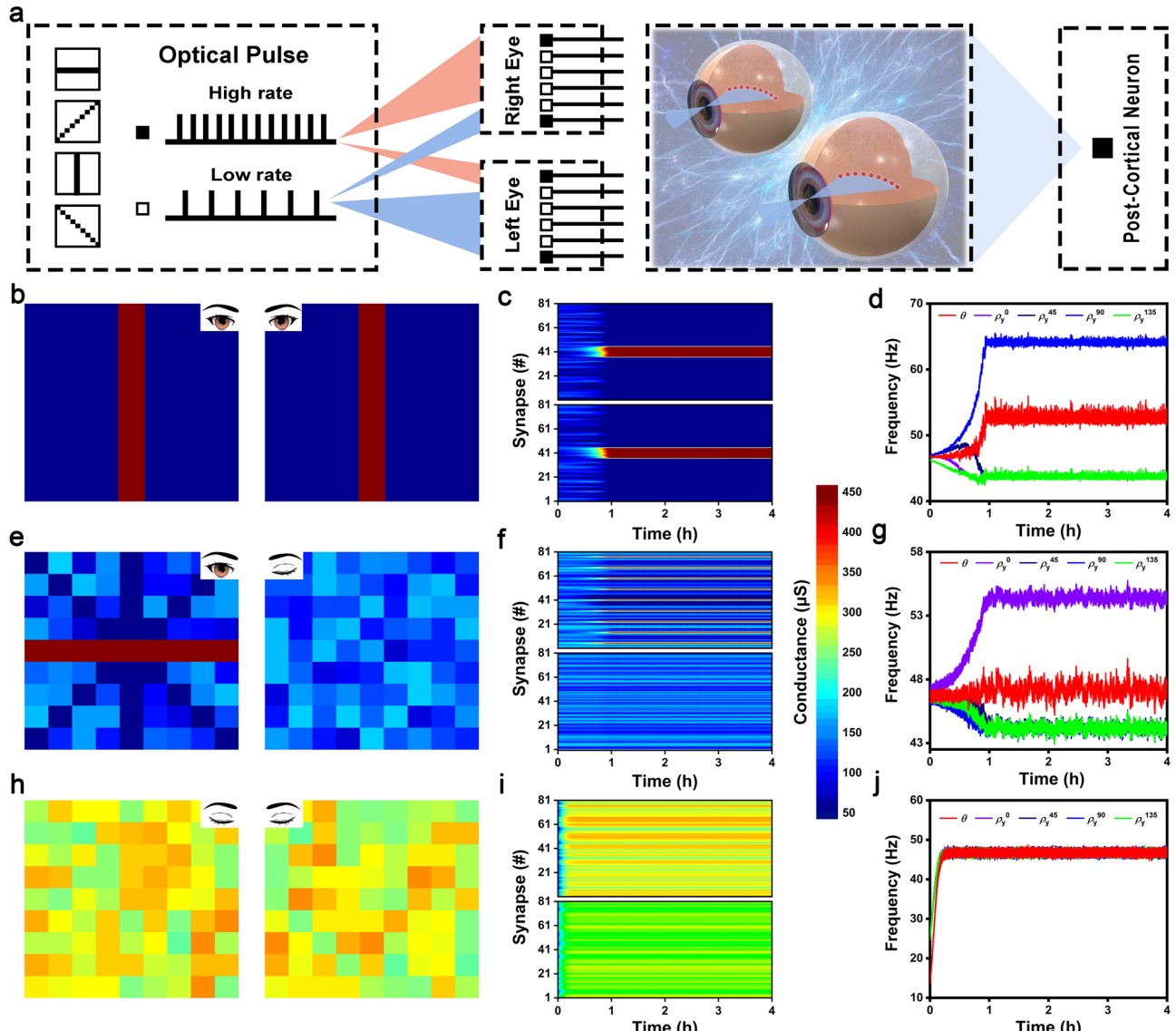

**Fig. 5 | Simulated large-scale crossbar array for artificial striate cortex with binocularity and orientation selectivity. a** Schematic of SNN for the orientation selectivity simulation. **b** The finial color maps of 9 × 9 cortical synapse arrays in normally rearing condition of orientation selectivity simulation, including two arrays for left and right eyes. **c** The evolution of synaptic weights as a function of simulation time, where the upper and lower behalf are corresponding to left and right eyes. **d** The evolution of postsynaptic firing frequencies and network threshold as a function of simulation time. It is clearly that only $\rho_y^{90}$ is larger than firing threshold which means the vertical is the selected orientation. **e–j** Similarly, the simulation results of monocular deprivation and binocular deprivation.

contour vision, (2) monocular deprivation, (3) binocular deprivation, as shown in Fig. 5b–j, respectively. The first two columns are the final maps of synaptic weights in left and right eyes, and the following two columns are the evolution of synaptic weights and postsynaptic firing rates/sliding threshold as a function of eclipsed time. In the normal binocular contour vision, the orientation inputs of left and right eyes are identical in every epoch. At the beginning of the simulation, the synaptic weights exhibit less variation, the four postsynaptic firing rates and sliding threshold have similar resultant values, without selectivity capacity. After a one-hour training, the vertical orientation has been randomly selected and continue potentiated with its post-synaptic firing rate gradually increasing to be larger than threshold ($\rho_y^{90}>\theta$). Meanwhile, the other orientations gradually perform with lower postsynaptic firing rate ($\rho_y^0, \rho_y^{45}, \rho_y^{135}<\theta$). Finally, the SNN network reaches a steady state after four-hour learning process with the vertical orientation as the winner orientation. In the monocular deprivation condition, we assume that the right eye is deprived, which is input with image containing noise pixels (randomly selected in the rage of 4 to 6 Hz). It is clearly that, the left eye network with normal input image develops a selection of horizontal orientation successfully while the right eye network loses its responsiveness and the synaptic weight always fluctuate in the low weight region. The postsynaptic firing rate of the winner horizontal orientation ($\rho_y^0$=54 Hz) is lower than that of the winner vertical orientation ($\rho_y^{90}$=64 Hz) in normal rearing simulation. The sliding threshold keeps greatly fluctuation even in the steady state of simulation. The degraded performance is attributed to the unrecognizable and negative effects of the noise input from right eye. In the binocular deprivation condition, both eyes are input with image containing noise pixels. Unsurprisingly, the orientation selectivity is lost. All the simulation results are in agreement with experimental findings, indicating a feasibility of our self-powered artificial cortex for developing the receptive field with binocularity and orientation selectivity[67].

## Discussion

In summary, based the crossbar array of self-powered memristors, we first emulated artificial striate cortex with binocularity, orientation selectivity based on the BCM learning rule. The crossbar array of self-powered memristors is monolithic all perovskite system where each cross point contains one CsFAPbI$_3$ perovskite solar cell (photosensory retinal cell) to convert external optical signals into electrical signals and one CsPbBr$_2$I perovskite memristor (cortical synapse) to implement plasticity modulating. Based on this artificial striate cortex, we investigated the triplet-STDP rules under optical stimuli. The asymmetry characteristic of triplet-STDP is beneficial to the following realization of BCM-rate learning rule compared to pair-STDP. By constructing the $3 \times 3$ crossbar array of self-powered memristor, the critical characteristics of BCM, synapse depression/potentiation takes place at low/high postsynaptic firing rate region, and the history-dependent sliding threshold were realized which has been further applied in the optical-encoded pattern recognition. Finally, artificial striate cortex with binocularity and orientation selectivity based on two simulated $9 \times 9$ self-powered memristor networks, following the generalized BCM learning rule. By varying the type of input for (1) normal binocular contour vision, (2) monocular deprivation, (3) binocular deprivation, we highly reproduced the experience-dependent modifications that have been observed experimentally in kitten striate cortex. Two-terminal structure of self-powered memristor based on monolithic all-perovskite system ensures the bio-inspired striate cortex to be extendable to crossbar array structure for high-density and low power consumption machine vision, which has not been realized yet.

## Methods

### Materials preparation

The CsPbBr$_2$I perovskite precursor was prepared by reacting PbI$_2$ (0.461 g, 99.99%), PbBr$_2$ (0.367 g, 99.999%) and CsBr (0.425 g, 99.9%) in the 4 mL component solvent of dimethyl sulfoxide (DMSO, 99.9%) and N,N-dimethylformamide (DMF, 99.8%) with V$_{DMSO}$:V$_{DMF}$ = 1:1. The reacting solution was stirred at ambient temperature (25 °C) for 30 minutes and then filtered the impurities. P3HT precursor was prepared by reacting P3HT in the trichloromethane (CHCl$_3$) as 3 mg/ml which followed by 1 hour dispersed ultrasonically and impurities filter processes. The 1.5 M CsFAPbI$_3$ perovskite precursor solution was prepared by dissolving FAI of 245.05 mg, CsI of 19.47 mg, PbI$_2$ of 681.13 mg, PbBr$_2$ of 8.53 mg, and MACl of 35 mg in the mixed solvents of DMF and DMSO (V$_{DMF}$:V$_{DMSO}$ = 4:1). The Spiro-OMeTAD (72.3 mg), 4-tert-butyl pyridine (tBP, 28.8 μL) and lithium bis (trifluoromethanesulfonyl) imide (Li-TFSI, 17.5 μL) stock solution (520 mg Li-TSFI in 1 mL acetonitrile) were dissolved in 1 mL chlorobenzene (CB), using as the Spiro-OMeTAD precursor.

### Perovskite memristor fabrication

The ITO substrates used in this work were sequentially treated by warm deionized water, acetone, isopropanol and UV-ozone. the CsPbBr$_2$I perovskite film was spin-coated on the ITO at 500 rpm for 5 s and 2000 rpm for 30 s, and 10 min/100 °C annealing process was also needed. Then, P3HT film deposited by spin coating at a spinning speed of 500 rpm for 5 s and at 3000 rpm for 10 s, followed by being annealed at 100°C for 10 min. Finally, the Au top electrodes were deposited by thermal evaporation through a metal shadow mask.

### Artificial striate cortex fabrication

We first fabricated solar cell by sandwiching a perovskite active layer between two selective charge transport layers and electrodes where SnO$_2$ is electron transport layer for negative charge extraction, spiro-OMeTAD film functions as hole transports layer for positive charge extraction, ITO is anode and Au electrode functions as cathode. The colloidal solution of SnO$_2$ (3.75 wt%) was spin-coated on the ITO substrates at 3000 rpm for 30 s and then the SnO$_2$-coated ITO substrates were annealed at 150 °C for 30 min. It's worth noting that the diluted SnO$_2$ colloidal solution was filter by the 0.45 μm PVDF filter before use. Then, the CsFAPbI$_3$ perovskite film was prepared by one-step spin coating method and followed an annealing process at 150 °C for 30 min. The filtered Spiro-OMeTAD solution by the 0.45 μm PTFE filter was spin coated onto the perovskite layer at 3000 r.p.m. for 30 s. The Au cathode were deposited by thermal evaporation through a metal shadow mask. Subsequently, we directly integrated memristor onto the as-fabricated solar cell by the following procedures: the 600 nm ITO layer were deposited onto the Au cathode of solar cell by magnetron sputtering which are compact enough to protect the as-formed sub-solar cell from damage during solution processing of perovskite memristor. Then, the Au electrodes were deposited on ITO by thermal evaporation through a metal shadow mask. Ultrathin P3HT film was spin-coated onto Au layer to function as reservoir layer to accept the migrated halide ions for continuous modulation of conductance. The CsPbBr$_2$I was spin-coated on the P3HT film to act as switching layer. Finally, the ITO was deposited by vacuum evaporation.

### Device and system characterization

The electrical properties of our memristor and photoelectric synapse were measured with a B1500A semiconductor characterization system at room temperature. The current density-voltage (J-V) characteristic of the solar cell was measured in a N$_2$ glove box using a Keithley 2400 Source Meter under standard AM1.5 G illumination (SS-F5; Enli Technology, Taiwan) which calibrated by a silicon reference cell under a light intensity of 100 mW/cm$^2$. The implementation of BCM learning rule can be achieved based on a rational triplet-STDP scheme by elaborate designing the fire time of post-synaptic pulses. For instance, two post-synaptic pulses evenly distribute on the two sides of pre-synaptic pulse and the post-synaptic firing rate can be calculated as the reciprocal value of interval time of two post-synaptic pulses. The transition of memristor conductance states is identified as the result of BCM learning rule with a designed post-synaptic firing rate. For the implementation of light pattern recognition, the array of artificial striate cortex was divided to two groups based on different frequency ranges of optical input signal. One group consists of five cells whose input is high-frequency optical signal and another group contains four cells whose input is low-frequency optical signal. The cells in the same group triggered by the same optical stimuli can be operated synchronously which greatly simplified our experimental setup. The optical signal with different frequency was generated by combining the normal light source with rotating disc optical chopper. UV-vis absorption spectra was measured on a UV-vis spectrometer (Shimadzu UV-1800, Japan). The steady-state PL spectra was obtained by a fluorescence spectrophotometer (Cary Eclipse, Agilent) with an excitation wavelength of 510 nm. The time-resolved PL measurements was carried out using a combined fluorescence lifetime and steady-state spectrometer (FLS980, Edinburgh Instruments Ltd.) by using a 510 nm picosecond pulsed laser. EIS measurement was performed on CHI760E Electrochemical Workstation (Chen Hua, China) in the frequency range of 1 MHz to 0.1 Hz under dark illumination. TPC and TPV measurements were performed with a system excited by a 520 nm (3nJ, 60 ns) pulse laser. The XRD measurements was performed on D8advance (Buker), the the scan range of $2\theta$ is from 10° to 50° with a step of 0.02°. The XPS characterization was carried out using 250xi (Thermo ESCALAB). The AFM (Bruker Dimension Icon) and SEM (SU8010, HITACHI) were used to detailed analyse the film morphology and cross sectional structure of our device.

## Data availability

The data that support the plots within this paper are available from the corresponding author upon reasonable request.

## Code availability

The code can be available from the corresponding author upon reasonable request.

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

## Acknowledgements

This research was supported by the NSFC Program (grant nos. 62122055, 62074104 and 61974093), Guangdong Provincial Department of Science and Technology (grant nos. 2018B030306028), the Science and Technology Innovation Commission of Shenzhen (grant nos. 20200804172625001).

## Author contributions

Y.R., X.B., Y.G., J.W. fabricated the artificial striate cortex, performed experimental studies in devices and circuits, analyzed the data. M.W., and G.L. helped with solar cell fabrication and data analysis. Y.Y. and M.Z. helped with devices integration. Y.Z. helped with devices characterization. Y.R. performed the simulation work and wrote the paper. S.T.H. conceived and supervised the project and finalized the paper. All authors discussed the results and revised the manuscript.

## Competing interests

The authors declare no competing interests.
