## [Peer Review File · Nature Communications]

Synaptic plasticity in self-powered artificial striate cortex for binocular orientation selectivityReviewer #1 (Remarks to the Author):

The paper presented by Ren et al. shows an interesting technological solution to reproduce the striate visual cortex with combination of photovoltaic and memristive element. There is no clear novelty in terms of physics or components and the clear contribution is at the integration level since photovoltaic cell and memristor are co-integrated into a crossbar array. They use both physical demonstration and simulation to emulate bio-inspired learning.

Neuromorphic engineering is clearly a hot topic with a large readership and the proposed work fits well in this direction. I have several concerns about the concept in itself and about the possibility to scale this technological solutions toward futur neuromorphic vision sensors.

MAJOR COMMENTS

1/ The physical description of the system presents some unclear aspects:

- The memristor technology used here is claimed to be a 2nd order memristor. 2nd order memristor needs to have two different regimes (i.e. short term and long term dynamics). Long term dynamics should be able to lead to non-volatile effects. It is not clear to me that such non-volatile mechanism can be achieved here. The two time constant (from 1 to 10 ms) is not showing any distincts STP/LTP regimes. The proposed memristor seems to be purely volatile (with relaxation time constant in the 10 ms range). A clear LTP regime needs to be demonstrated

- Figure 2b is not clear: it is not possible to understand the switching principles since there is no loops orientation (SET and RESET needs to be identified). From the shape of the curves, negative polarity is a SET and positive polarity is a RESET, but labels and other figures in the manuscript are pointing in the other directions. Please clarify. (in the following, I will consider that the pulse programming polarity is correct, not the sweeping loops)

2/ neuromorphic functions

- The switching seems to be bipolar for this devices. Positive polarity induces potentiation and negative polarity induces depression (fig 2). Optical pulses corresponds to positive voltage and induces potentiation. Feedback electrical pulses corresponds to negative polarity and should induce depression. In order to get some BCM behavior, one expect to get a multiplicative effect of ρ_x and ρ_y on conductance. In other words, ρ_x alone should not lead to change of conductance and ρ_y alone neither (this is the intrinsic idea of learning). With the proposed switching schemes, I don't see how it is possible. A baseline experiment with pre_only and post_only would help to clarify. It looks like the BCM aspect is mostly described by the natural relaxation of the device that tends to reach its stable state (this issue is to be discussed in the light of the non-volatile question).

- the interaction between pre and post pulses is not clear, a better description is required

- The BCM fitting with experimental data is not clear. Additionally, there is several simplification that are difficult to follow. How do justify $A3 = 0$? $A2 \rightarrow A2 \cdot \rho_y^2 / \rho_0^2$?

3/ System level integration.

- there is no details about the design of the memristor and photovoltaic cells. Some microscopic images of the crossbar would be very beneficial. A related concerns is how compatible are photovoltaic cells and memristors in terms of power. Switching power for memristors is known to be a limitation since it requires "high" power. Is a scaled photovoltaic cell (with equivalent dimensions) could provide such power?

ADDITIONAL COMMENTS:

- Synapse is not a verb

- L84: crossbar paradigm only shows high density. There is no power consumption associated to this integration scheme

- fig 2b: it looks like it is absolute value of current. This is not mentioned (and not useful).

- L293: you don't demonstrate the exact implementation of BCM.

- Fig 5h: why mean frequency start at 10 Hz why figure d and g start at 48Hz?

- Fig 4f: how could you get a 400% change of conductance? All previous experiments

are showing a change of 40 %.

- Not clear what is ρ_y : sum of all ρ_x or is it a sequential experiment on each pixel? Is it Poisson distributed pulses? How do you manage synchronization issue if not?

Reviewer #2 (Remarks to the Author):

In this paper, the authors reported the bioinspired striate cortex with binocular and orientation-selective receptive field with the crossbar array of self-powered memristors. Each cross-point contained a perovskite solar cell directly stacking on a perovskite memristor. Furthermore, the plasticity of the flexible crossbar array of self-powered memristor was modulated with a generalized BCM learning rule for optical-encoded pattern recognition. This manuscript is well organized and contains technical advancements in this field. However, the experimental results in this manuscript are not enough to support the authors' claim. Authors need to improve the quality of the manuscript.

1. The visual information is supplied to the device continuously. Therefore, the artificial striate cortex would be exposed to excessive inputs. To overcome these environments, the device should have high endurance characteristics. The authors need to provide additional data about the electrical endurance characteristics of the device and array.

2. According to the manuscript, the authors utilized the perovskite material (CsPbBr_2I) for the memristor. Therefore, the memristor would have light reactivity when the optical inputs were applied to the device for the operation. The authors need to confirm that the self-powered devices were irrelevant to the light reactivity of the memristor. Also, the authors should provide additional explanations about the effect of the optical responses of the memristor during the device/array operation.

3. In Fig. 4, the size of the flexible array seemed to be large. What is the dimension of the array? Also, the presented array may need a large surface area to receive sufficient visual information from the environment whereas a small device size would be required to achieve the high-integration array for the recognition performance. Is there any plan to overcome these conflicting characteristics?

4. According to the manuscripts, the authors demonstrated the flexible array, which was the hemisphere shape. The authors need to provide the characteristics of the device/array depending on the different bending states.

5. In Fig. 3d, the authors showed the current response as a function of different light intensities. Is it possible to simulate the device/array with the function of light intensity? Also, is it possible to recognize the stripe patterns through simulation even when the light intensity and color map are mixed?

6. According to the manuscript, the authors claimed that they emulated a binocular orientation selectivity. The main function of binocular vision in a bio-system is to determine the distance to the object through the phase difference between two eyes. On the other hand, binocular characteristics are not required to recognize the striped pattern. The simulated result of Fig. 5 was seemed to increase of recognition rate as the number of arrays increased rather than the effect of the binocular orientation (The higher recognition performance would be obtained from the number of devices or arrays other than effect of the binocular orientation.). Therefore, it seems difficult to express these characteristics as binocular functions.

7. According to the manuscript, the flexible array was presented in the form of a hemisphere. However, in such a structure, distortion could exist when recognizing straight objects. Did this structure and distortion affect the recognition of striped patterns? Also, is this reflected in the simulation? The authors need to provide additional data and explanations.

Reviewer #3 (Remarks to the Author):

In this work, the authors fabricate a self-powered memristor array powered by a directly integrated solar-cell which makes the memristor can be modulated by the external light pulses just like photosensory neurons and synapses in the human retina. The proposed memristor also has a second-order characteristics, making it possible to control the memristor with non-overlapping pulses in STDP or SRDP learning rule, which enables bio-plausible memristor based neuromorphic system. Based on the self-powered and second-order effect of the memristor-solar cell integrated array, the authors demonstrated the orientation selectivity in binocular condition with BCM learning rule. The novelties of this manuscript are 1) BCM learning rule from self-powered memristor array and 2) binocular orientation selectivity using the memristor based system. However, the first novelty is already discussed from the authors' lab (ref. 40), and BCM learning rule using memristors has been proved by others work (ex. Ref. 50). The second novelty is valid while monocular orientation selectivity has been discussed in others work (ex. 50)

In other words, previous studies such as reference 40 and 50 in the paper already demonstrated the self-power memristor by directly integrating a solar cell and a memristor, and the monocular orientation selectivity with BCM learning rule based on the second-order memristors, but the orientation selectivity experiment based on the self-powered memristor has an advantage in that the optical signal directly affects the synapse similar to the photosensory neuron in the human's visual system. The integration of the memristor device and the solar cell is interesting, and the second-order effect of the memristor is highly desirable for a bio-plausible applications, however, there are several points that the authors should provide more detailed explanations and they should clarify the significance of the work (binocular orientation selectivity) compared to the previous works, considering that the three of advantages in the paper i) binocular orientation selectivity, ii) BCM learning rule in second order memristor, and iii) self-powered memristor by integrating the solar cell and the memristor device have been demonstrated already. Please see below for more detailed comments.

1) The authors claim second order phenomena because the device has two time constants: one is for long-term steady state and the other is for short-term volatile state. However, the authors focused on the short-term volatile state and did not explain the behavior of long-term state. The main part of this manuscript is "memristor" device. The authors should provide "basic information" about memristors such as set voltage, reset voltage, endurance, retention, on/off ratio and spatio-temporal distributions. Without that information, the device cannot be verified. what is the condition of the short-term to long-term transition of the device, and how long the device retain the conductance state in the long-term and the short-term mode?

2) Usually, perovskite based memristors have very "short life time", which means the device characteristics are degraded in a few days. The authors should claim the life time of the device and why they chose perovskite materials for this application.

3) perovskite patterning: The authors did not show the real image of the array, even the single cell image. It is well known that it is very hard to form crossbar structure with perovskite due to fabrication issue. The authors should provide the real image of the array and stand-alone device. The information about device size needs to be provided, too.

4) Sneak path problem : How did the authors avoid the sneak path problem in memristor array? The I-V curve from figure 2b does not show any rectification.

5) The authors should provide information about solar cell and memristor devices such as solar cell output voltage, memristor input voltage range, etc.

6) no explanation about figure 1d in the manuscript.

7) The authors should provide the uniformity value from pulse measurement. If the variation is large, the figures (for example figure 4d) cannot be valid.

8) Please specify the figure number of supporting information in the manuscript. The authors just mentioned supporting information without any specific figure numbers.

9) What did the authors want to show from figure 3e and 3f? It is natural that the device has negative current with post-pulse and positive current with pre-pulse.

10) In 3 x 3 array test, how did the authors select specific cell for the light pattern? From the system, the light emits from outside and then all devices might receive the same light.

11) With the scheme in figure 4d, the sliding threshold effect is just a consequence of the previous history, and it is not a controllable parameter. And this behavior is common in general volatile memristors. The authors should comment how to control this sliding threshold effect. For example, the homeostasis, which is for controlling the number of spikes, can be controlled by gate voltage in 3 terminal devices. Also, is there any improvement by this sliding threshold effect on your application? The authors should comment on this, too.

12) The authors performed simulation by using 9 by 9 array. 9 by 9 array can be achieved experimentally. The authors should explain why larger array size was not utilized for the simulation. Also, the simulation seems that only one orientation was randomly learned. Are the other three orientations learned with similar probability?

13) In line 302, the SNN is trained with winner-take-all method, which usually represents a biological phenomenon that the first firing neuron inhibits the other neurons in the same layer to effectively reduce the unnecessary firing event and prevent learning of wrong information. In the paper, it is written that the winner-take-all method is applied among four different firing rates in a same post-synaptic neuron. It is confusing that how the winner-take-all method is applied to the given system, where there is only a single output neuron.

14) Because the self-power memristor characteristics, the demonstration of the sliding effect in BCM learning rule in memristors, and the orientation selectivity in the memristor based SNN are already studied, the strongest novelty in the paper is in the binocular orientation selectivity based on the self-power memristor, that can directly utilize the optical signal. However, it is not clear that what is the advantage of the binocular condition in the orientation selectivity. The authors claimed that the simulation results are well matched with experimental findings in line 326, but it would be better to emphasize the advantage of the demonstration in binocular condition.

15) Why was the 180 degree flexible structure made? Is it simply to imitate the shape of the eye? Are there any advantages to making a 180 degree flexible structure?

This paper is not carefully written. Please find the minor suggestions and corrections below.

1) Some sentences in main text are just copied and pasted from the abstract. The first sentence of the abstract is grammatically wrong.

2) 1) in line 75, synapse ==> synapses

3) in line 87, the round bracket is missing.

4) in line 320, wight ==> typo.

5) The sentence in line 110~111 needs to be correct.

6) In line 122 in supporting information, a resistor cannot have the unit "us".

Response to Reviewers' Comments

Reviewer's Comment

Our Response

Changes Made in the Manuscript

Reviewer #1 (Remarks to the Author)

Comments to the Author

The paper presented by Ren et al. shows an interesting technological solution to reproduce the striate visual cortex with combination of photovoltaic and memristive element. There is no clear novelty in terms of physics or components and the clear contribution is at the integration level since photovoltaic cell and memristor are co-integrated into a crossbar array. They use both physical demonstration and simulation to emulate bio-inspired learning. Neuromorphic engineering is clearly a hot topic with a large readership and the proposed work fits well in this direction. I have several concerns about the concept in itself and about the possibility to scale this technological solution toward future neuromorphic vision sensors.

We are glad that the reviewer finds this work fitting well in hot topic of neuromorphic engineering with a large readership. We would like to thank the reviewer for taking the time to read the manuscript and provide insightful feedback that has helped us greatly in improving the quality of the manuscript. We have provided point-by-point answers to the comments and concerns raised by the reviewer below. We have also highlighted the corresponding changes made to the manuscript.

The differences and novelty of our work are highlighted as below:

(1) Binocular orientation selectivity: In the biological visual system, both left and the right eyes receive the outside information and output signals to the striate cortex, V1 for first converge. Neurons in adult striate cortex are binocular with a strong selectivity for a particular orientation. For realizing normal binocular perceptions including stereopsis and depth, binocularly matched orientation selectivity between the two eyes is highly demanded. Binocular neurons in the striate cortex should match their orientation by tuning through the two eyes to perceive

coherently. As pointed out by the reviewer, reference 50 reported emulation of monocular orientation selectivity with BCM learning rule based on the second-order memristors (only functioning as synapse) without considering the associated neuronal activity between the two eyes to individual cortical neurons. In our work, the hardware implementation of artificial striate cortical neuron with binocular orientation selectivity based on self-powered memristor (combining the functions of both sensory neuron, cortical synapse and striate cortical neuron) has been first reported.

(2) Solution-processed monolithic all-perovskite system for self-powered artificial striate cortex: our previous work (reference 40) just demonstrated the self-powered artificial retina system by wire-bonding the silicon solar cell with perovskite memristor where the solar cell and memristor are separated. Specific interconnections between solar cell and memristor with inherent heterotypic materials restricts the simplifying process complexity and reducing packing density. However, in this work, the crossbar array of artificial striate cortex is the solution-processed monolithic all-perovskite system where each cross-point contains one CsFAPbI₃ perovskite solar cell directly stacking on the CsPbBr₂I perovskite memristor. Solution-processable perovskites are promising for low-cost, high-efficiency solar cells owing to their tunable bandgaps, high optical absorption coefficient, simple processing and high defect tolerance. In addition, benefiting from excellent ionic dynamic properties, perovskites are suitable for developing memristors with adjustable conductance properties and low power consumption. It remains challenging to fabricate solution-processed monolithic all-perovskite system for striate cortex since solution processing of a perovskite layer often induces damage to the underlying pre-formed perovskite films. It is also challenging to optimize the charge transport interface to ensure that solar cell could supply sufficient power to drive memristor. **By delicately compound engineering and structural design, we first realize solution-processed monolithic all-perovskite system to implement the hardware-based striate cortex with binocular and orientation selective receptive field.**

In summary, our work first reported hardware implemented striate cortex based on solution-processed monolithic all-perovskite system with binocular and orientation selectivity, as far as we understand. The bio-inspired striate cortex is highly compatible with high-density and low power consumption machine vision owing to its crossbar paradigm and homotypic

materials system.

1. The physical description of the system presents some unclear aspects:

(1.1) The memristor technology used here is claimed to be a 2nd order memristor. 2nd order memristor needs to have two different regimes (i.e. short term and long term dynamics). Long term dynamics should be able to lead to non-volatile effects. It is not clear to me that such non-volatile mechanism can be achieved here. The two time constant (from 1 to 10 ms) is not showing any distinct STP/LTP regimes. The proposed memristor seems to be purely volatile (with relaxation time constant in the 10 ms range). A clear LTP regime needs to be demonstrated.

We thank the reviewer for this constructive question. As correctly pointed by the reviewer, the second order memristor should exhibit 1st-order long-term variable modulation and 2nd-order short-term variable modulation, which are analogue to the long-term and short-term dynamics phenomenon in the biological synapse. As shown in the Fig. 2e and Fig. 2f, the long-term dynamics governed non-volatile characteristic of our memristor has been demonstrated where the continuously increased conductance states can be maintained after application of five pulses. To further verify the co-existence of short-term and long-term dynamics of memristor, we performed a series of pulse experiments on the memristor with the pulse number ranging from one to five (Fig. R1). The conductance of memristor decays to original state of 110 μ S after application of one single pulse while after application of five pulses, the conductance can be stabilized at 150 μ S for 10⁴ s, indicating the transition from short-term dynamics to long-term dynamics.

In addition, we re-fit PPF curve of memristor to obtain the STP time constants (τ_1) of 0.28 ms and LTP time constants (τ_2) of 10.86 ms (Table R1). Apparently, τ_2 is about one order of magnitude larger than τ_1 , which agrees well with the measured data in biological synapses (~ms order) and other artificial synapse works (Adv. Electron. Mater. 2019, 5, 1900287; Nat. Electron. 2021, 4, 348; Nat. Mater. 2016, 16, 101). In addition, the τ_1 and τ_2 are the time constants of relaxation processes, the states can be stabilized for 10⁴ s indicating the non-volatile characteristic of our memristor.

Following discussion was added in the supporting information or revised manuscript:

“In the psychological model of biology memory system, the perceptive information is firstly stored for a very short period of time in the sensory memory, and then the information can be transformed from short-term memory (STM) to long-term memory (LTM) through a rehearsal process, corresponding to the transition from short-term dynamics to long-term dynamics in neuroscience. To mimic this memory process, we monitored the current states by application of pulse trains ranging from one single pulse to five pulses where the pulse amplitude and width are set to 0.5V and 5ms, respectively. The conductance of memristor decays to original state of $110\ \mu\text{S}$ after application of one single pulse but can be stabilized at $150\ \mu\text{S}$ for $10^4\ \text{s}$ after application of five pulses, indicating the transition from short-term dynamics to long-term dynamics. “

“By fitting the PPF curve, the τ_1 of STP constant and τ_2 of LTP constant were estimated as 0.28 ms and 10.86 ms, respectively. The τ_2 is one order larger than τ_1 which is comparable to the biological synapse ($\sim\text{ms}$ order). “

Figure R1 a. The biological memory model proposed by Atkinson and Shiffrin. b. The memristor current responses to pulse train stimulations with different pulse numbers.

Parameter	c_1	τ_1	c_2	τ_2
Value	0.7547	0.28 ms	0.4663	10.86 ms

Table R1. The fitting results of PPF.

(1.2) Figure 2b is not clear: it is not possible to understand the switching principles since there is no loops orientation (SET and RESET needs to be identified). From the shape of the curves, negative polarity is a SET and positive polarity is a RESET, but labels and other figures in the manuscript are pointing in the other directions. Please clarify. (in the following, I will consider that the pulse programming polarity is correct, not the sweeping loops)

We would like to thank the reviewer for pointing out this problem. Note that Fig. 2b has been revised to include the loop orientation in the SET and RESET operations. We also recorded the transient current response as the function of time by application of five positive sweep from 0 V to 0.5 V to 0 V and subsequent five negative sweep from 0 V to -0.5 V to 0 V (Fig. R2). It is clearly that the SET and RESET can be triggered by positive and negative sweeps, respectively which is consistent with the pulse programming polarity.

Following discussion was added in the revised manuscript:

“The direct current (DC) measurement of memristor shows the pinched hysteresis phenomenon, as shown in Fig. 2b. Different from the bistable resistive switching phenomenon, the response current of our device gradually increases during application of five consecutive positive sweeps and decreases followed five consecutive negative sweeps, indicating tunable conductance characteristics which is analogue to the modulation of biological synapses.”

Figure R2. The current with respect to DC voltage sweeps, the device conductance gradually increase (decrease) under positive (negative) sweeps of 0.5 V (-0.5 V).

2. Neuromorphic functions

(2.1) The switching seems to be bipolar for this device. Positive polarity induces potentiation and negative polarity induces depression (Fig. 2). Optical pulses correspond to positive voltage and induces potentiation. Feedback electrical pulses corresponds to negative polarity and should induce depression. In order to get some BCM behavior, one expects to get a multiplicative effect of ρ_x and ρ_y on conductance. In other words, ρ_x alone should not lead to change of conductance and ρ_y alone neither (this is the intrinsic idea of learning). With the proposed switching schemes, I don't see how it is possible. A baseline experiment with `pre_only` and `post_only` would help to clarify. It looks like the BCM aspect is mostly described by the natural relaxation of the device that tends to reach its stable state (this issue is to be discussed in the light of the non-volatile question).

This is an excellent question raised by the reviewer. We believe that there is a misunderstanding between the BCM learning rules and traditional synaptic plasticity emulation. Note that the

change of conductance (synaptic weight) based on BCM learning rule shows multiplicative effect of ρ_x and ρ_y . However, one cannot execute this modeling by directly setting $\rho_x = 0$ or $\rho_y = 0$ since it is inconsistent with the definition of BCM in its biological counterpart. First, the BCM learning rule describes the synaptic modulation of the synergistic effect of ρ_x and ρ_y which require to be stimulated synchronously. Second, the synapses constantly receive and transmit the signals, and the $\rho_x = 0$ or $\rho_y = 0$ is irrational in the biological system even under resting-state. As suggested by reviewer, we have performed control experiment with $\rho_x = 0$ or $\rho_y = 0$ as shown in Fig. 2d, the memristor conductance can be gradually increased by consecutive 100 pre-synaptic spikes (without post-synaptic spikes, $\rho_y = 0$) and decreased by 100 post-synaptic spikes (without pre-synaptic spikes, $\rho_x = 0$), which is corresponding to the synaptic long-term potentiation (LTP) and long-term depression (LTD) behaviors. This control experiments demonstrate that the memristor possesses programmable conductance levels which is suitable for synaptic emulation. However, for implementing BCM learning rules, as discussed above, either $\rho_x = 0$ or $\rho_y = 0$ is irrational.

(2.2) The interaction between pre and post pulses is not clear, a better description is required

We would like to thank the reviewer for pointing out this problem. We clarified the interaction between pre- and post- pulses as below: in the pattern learning and the simulation works, the pre-synaptic pulses were applied with the rate of $\rho_x = [\rho_x^1, \rho_x^2 \cdots \rho_x^i]$, where the i is the index of the synapse. The post-synaptic pulses fired following a linear relationship with the rate of $\rho_y = G\rho_x = \sum_{i=1}^9 G_i \rho_x^i$, where $G = [G_1, G_2 \cdots G_i]$ is the weights of the synaptic array.

Following discussion was added in the revised manuscript:

“The post-cortical neuron responds to the input singles following a linear function with a firing

rate of $\rho_y = G\rho_x = \sum_{i=1}^9 G_i \rho_x^i$, where $G = [G_1, G_2 \dots G_i]$ and $\rho_x = [\rho_x^1, \rho_x^2 \dots \rho_x^i]$ (i is the index of the synapse) are the synapse weight and input rate, respectively.”

(2.3) The BCM fitting with experimental data is not clear. Additionally, there is several simplifications that are difficult to follow. How do justify $A_3^- = 0$? $A_2^- \rightarrow A_2^- \cdot \rho_y^2 / \rho_y^0$?

We would like to thank the reviewer for pointing out this problem. We clarified BCM fitting with experimental data as below: In the fitting process of BCM learning rule, several simplifications have been carried out to emulate the BCM features. As mentioned in original manuscript, the BCM learning rule can be described based on the All-to-All framework as $dG/dt = \phi(\rho_y, \theta) \rho_x = (-A_2^- \tau_- \rho_y - A_3^- \tau_- \tau_x \rho_x \rho_y + A_2^+ \tau_+ \rho_y + A_3^+ \tau_+ \tau_y \rho_y^2) \rho_x$. The first simplification is to view A_3^- as zero. For the requirement of mathematic fitting, the $\phi(\rho_y, \theta) = -A_2^- \tau_- \rho_y - A_3^- \tau_- \tau_x \rho_x \rho_y + A_2^+ \tau_+ \rho_y + A_3^+ \tau_+ \tau_y \rho_y^2$ is designed as scalar function to emulate the essential feature of BCM learning rule: the high post-synaptic firing frequency corresponding to synaptic potentiation ($(\rho_y > \theta) > 0$), the low post-synaptic firing frequency corresponding to synaptic depression ($(\rho_y < \theta) < 0$) and the synapse maintaining its state when post-synapse doesn't fire ($(0, \theta) = 0$). According to the biological experimental results, the BCM learning rule for synaptic modification in biological system obey a zero-axial quadratic polynomial with the argument of ρ_y so that setting A_3^- to zero is on-demand. In addition, the A_3^- is much lower than A_2^- ($A_3^- \ll A_2^-$) in biological system which means the effect of the triplet depression term is almost negligible compared with the depression induced by paired spikes. Hence, this simplification will not affect the emulation of BCM learning in biological system.

Even the previous simplification of $\phi(\rho_y, \theta)$ is capable of emulating the first feature of BCM learning rule, another simplification it still required to mimic the second feature, namely, the sliding threshold. The average value of second power of postsynaptic firing rate $\langle \rho_y^2 \rangle$, which

reflects previous history of memristor, is introduced to describe the sliding feature of threshold. The first order term of $\phi(\rho_y, \theta)$ is selected and redefined as $A_2^- \rightarrow A_2^- \langle \rho_y^2 \rangle / \rho_0^2$ and $A_2^+ \rightarrow A_2^+ \langle \rho_y^2 \rangle / \rho_0^2$, while the second order term is kept unchanged. In conclusion, these simplifications allow us to implement a rational BCM model with relatively less parameters.

Following discussions was included in the supporting information:

“In the fitting process of BCM learning rule, several simplifications have been carried out to emulate the BCM features. As mentioned in original manuscript, the BCM learning rule can be described based on the All-to-All framework as $dG/dt = \phi(\rho_y, \theta) \rho_x = (-A_2^- \tau_- \rho_y - A_3^- \tau_- \tau_x \rho_x \rho_y + A_2^+ \tau_+ \rho_y + A_3^+ \tau_+ \tau_y \rho_y^2) \rho_x$. The first simplification is to view A_3^- as zero. For the requirement of mathematic fitting, the $\phi(\rho_y, \theta) = -A_2^- \tau_- \rho_y - A_3^- \tau_- \tau_x \rho_x \rho_y + A_2^+ \tau_+ \rho_y + A_3^+ \tau_+ \tau_y \rho_y^2$ is designed as scalar function to emulate the essential feature of BCM learning rule: the high post-synaptic firing frequency corresponding to synaptic potentiation ($(\rho_y > \theta, \theta) > 0$), the low post-synaptic firing frequency corresponding to synaptic depression ($(\rho_y < \theta, \theta) < 0$) and the synapse maintaining its state when post-synapse doesn't fire ($(0, \theta) = 0$). According to the biological experimental results, the BCM learning rule for synaptic modification in biological system obey a zero-axial quadratic polynomial with the argument of ρ_y so that setting A_3^- to zero is on-demand. In addition, the A_3^- is much lower than A_2^- ($A_3^- \ll A_2^-$) in biological system which means the effect of the triplet depression term is almost negligible compared with the depression induced by paired spikes. Hence, this simplification will not affect the rationality of emulation of BCM learning in biological system.

Even the previous simplification of $\phi(\rho_y, \theta)$ is capable of emulating the first feature of BCM learning rule, another simplification it still required to mimic the second feature, namely, the sliding threshold. The average value of second power of postsynaptic firing rate $\langle \rho_y^2 \rangle$, which reflects previous history of memristor, is introduced to describe the sliding feature of threshold.

The first order term of $\phi(\rho_y, \theta)$ is selected and redefined as $A_2^- \rightarrow A_2^- \langle \rho_y^2 \rangle / \rho_0^2$ and $A_2^+ \rightarrow A_2^+ \langle \rho_y^2 \rangle / \rho_0^2$, while the second order term is kept unchanged. In conclusion, these simplifications allow us to implement a rational BCM model with relatively less parameters.”

3. System level integration: there is no detail about the design of the memristor and photovoltaic cells. Some microscopic images of the crossbar would be very beneficial. A related concerns is how compatible are photovoltaic cells and memristors in terms of power. Switching power for memristors is known to be a limitation since it requires "high" power. Is a scaled photovoltaic cell (with equivalent dimensions) could provide such power?

We thank the reviewer for this constructive question. In this work, the crossbar array of artificial striate cortex is the solution-processed monolithic all-perovskite system where each cross-point contains one CsFAPbI₃ perovskite solar cell directly stacking on the CsPbBr₂I perovskite memristor. Solution-processable metal halide perovskites are attractive for high-efficiency, low-cost solar cells owing to their tunable bandgaps, high optical absorption coefficient, simple processing and high defect tolerance. In addition, benefiting from excellent ionic dynamic properties, perovskites are suitable for developing memristors with adjustable conductance properties and low power consumption. Therefore, the key idea of our design is to employ solution-processed monolithic all-perovskite system to implement the hardware-based striate cortex with binocular and orientation selective receptive field.

Reviewer’s concern regarding compatibility of photovoltaic cells and memristors in terms of integration and power is valid. It remains challenging to fabricate solution-processed monolithic all-perovskite system for striate cortex since solution processing of a perovskite film often induces damage to the underlying as-formed perovskite layers. It is also challenging to optimize the charge transport interface to ensure that solar cell could supply sufficient power to drive memristor.

The details about the design of the memristor and photovoltaic cell are as below: we first fabricated solar cell by sandwiching a CsFAPbI₃ perovskite active layer between two selective charge transport layers and electrodes where SnO₂ is electron transport layer for negative charge

extraction, spiro-OMeTAD film functions as hole transports layer for positive charge extraction, ITO is anode and Au electrode functions as cathode. Subsequently, we directly integrated memristor onto the as-fabricated solar cell by the following procedures: the 600 nm ITO layer were deposited onto the Au cathode of solar cell by magnetron sputtering which are compact enough to protect the as-formed sub-solar cell from damage during solution processing of perovskite memristor. Then, the Au top electrodes were deposited on ITO by thermal evaporation through a metal shadow mask. Ultrathin P3HT film was spin-coated onto Au layer to function as reservoir layer to accept the migrated halide ions for continuous modulation of conductance. The CsPbBr₂I was spin-coated on the P3HT film to act as switching layer. Finally, the ITO BE was deposited by vacuum evaporation.

The dimension of the integrated artificial striate cortex array is 3 cm × 3 cm with the electrode width of 1000 μm. Each cross-point is an artificial striate cortex cell with the structure of ITO/CsPbBr₂I/P3HT/Au/ITO/Au/Spiro/CsFAPbI₃/SnO₂/ITO. The array size is limited by the size of sub-solar cell for receiving sufficient visual information from the environment. The microscopic images were obtained by performing the cross-sectional scanning electron microscopy (SEM) on device where the continuous and smooth surface of each layer can be observed, verifying that our design of solution-processed monolithic all-perovskite system allows the facile integration of memristor and solar cell without damaging of each layer.

Figure R3. The illustration of artificial striate cortex array and the cross-sectional SEM image of the single cell.

Regarding compatibility of photovoltaic cells and memristors in terms of power, we performed

pulse experiment on the memristor by application of single pulse with amplitude of 0.5 V and duration of 5 ms (Fig. R4). The power consumption of single memristor per operation is estimated as $P_{\text{potentiation}} = V \times I \times t = 0.5 \text{ V} \times 70 \text{ } \mu\text{A} \times 5\text{ms} = 175 \text{ nJ}$. In addition, more detailed parameters about solar cell used in this work under 1-Sun illumination are provided in Table R2. The solar cell exhibits max power of 0.43 mW, the V_{oc} of 1.1 V, and J_{sc} of 23.7 mA/cm², indicating that the solar cell could generate enough power to drive the memristor.

Following discussion was added in the supporting information or revised manuscript:

“The microscopic images were obtained by performing the cross-sectional scanning electron microscopy (SEM) on device where the continuous and smooth surface of each layer can be observed, verifying that our design of solution-processed monolithic all-perovskite system allows the facile integration of memristor and solar cell without damaging of each layer.”

“And the dimension of the array is 3 cm × 3 cm with the electrode width of 1000 μm, where the large array size is limited by the size of sub-solar cell for receiving sufficient visual information from the environment.”

“Regarding compatibility of photovoltaic cells and memristors in terms of power, we performed pulse experiment on the memristor by application of single pulse with amplitude of 0.5 V and duration of 5 ms. The power consumption of single memristor per operation is estimated as $P_{\text{potentiation}} = V \times I \times t = 0.5 \text{ V} \times 70 \text{ } \mu\text{A} \times 5\text{ms} = 175 \text{ nJ}$. In addition, more detailed parameters about solar cell used in this work under 1-Sun illumination are provided in Table S3. The solar cell exhibits max power of 0.43 mW, the V_{oc} of 1.1 V, and J_{sc} of 23.7 mA/cm², indicating that the solar cell could generate enough power to drive the memristor.”

“We first fabricated solar cell by sandwiching a perovskite active layer between two selective charge transport layers and electrodes where SnO₂ is electron transport layer for negative charge extraction, spiro-OMeTAD film functions as hole transports layer for positive charge extraction, ITO is anode and Au electrode functions as cathode. The colloidal solution of SnO₂ (3.75 wt%) was spin-coated on the ITO substrates at 3000 rpm for 30 s and then the SnO₂-coated ITO substrates were annealed at 150°C for 30 min. It's worth noting that the diluted SnO₂ colloidal solution was filter by the 0.45 μm PVDF filter before use. Then, the CsFAPbI₃ perovskite film was prepared by one-step spin coating method and followed an annealing

process at 150°C for 30 min. The filtered Spiro-OMeTAD solution by the $0.45\ \mu\text{m}$ PTFE filter was spin coated onto the perovskite layer at 3,000 r.p.m. for 30 s. The Au cathode were deposited by thermal evaporation through a metal shadow mask. Subsequently, we directly integrated memristor onto the as-fabricated solar cell by the following procedures: the 600 nm ITO layer were deposited onto the Au cathode of solar cell by magnetron sputtering which are compact enough to protect the as-formed sub-solar cell from damage during solution processing of perovskite memristor. Then, the Au top electrodes were deposited on ITO by thermal evaporation through a metal shadow mask. Ultrathin P3HT film was spin-coated onto Au layer to function as reservoir layer to accept the migrated halide ions for continuous modulation of conductance. The CsPbBr_2I was spin-coated on the P3HT film to act as switching layer. Finally, the ITO BE was deposited by vacuum evaporation.”

Parameter	Area	I_{sc}	V_{oc}	P_{max}	I_{max}	V_{max}	Efficiency	Fill Factor	J_{sc}	J_{max}
Value	0.02cm^2	0.47mA	1.1V	0.43mW	0.45mA	0.95V	21.6%	82.6%	23.7 mA/cm ²	22.7 mA/cm ²

Table R2. The detailed parameters of the solar cell employed in this work.

Figure R4. The current response of memristor under external voltage pulse with amplitude of 0.5 V and duration of 5 ms.

4. ADDITIONNAL COMMENTS:

(4.1) Synapse is not a verb

We would like to thank the reviewer for pointing out this error. We have revised it in the manuscript.

(4.2) L84: crossbar paradigm only shows high density. There is no power consumption associated to this integration scheme

In this work, the width of both optical pulse and electrical pulse are set to 5 ms, and the memristor current is selected as $70\mu\text{A}$, as shown in Fig. R4. For the memristor potentiation, the photovoltage of solar cell is 1 V, hence, the power consumption of the single potentiation operation is about $P_{\text{potentiation}} = V \times I \times t = 0.5 \text{ V} \times 70\mu\text{A} \times 5 \text{ ms} = 175 \text{ nJ}$. While for the single depression operation, the voltage pulse is directly applied on the memristor with amplitude of 0.5 V, hence, the $W_{\text{depression}}$ is estimated as $V \times I \times t = 0.5 \text{ V} \times 70\mu\text{A} \times 5 \text{ ms} = 175 \text{ nJ}$.

(4.3) Fig 2b: it looks like it is absolute value of current. This is not mentioned (and not useful).

We would like to thank the reviewer for pointing out this error. We recorded the transient current response as the function of voltage sweep to replace the Fig. 2b in the revised manuscript:

Following discussion was added in the revised manuscript:

“The direct current (DC) measurement of memristor shows the pinched hysteresis phenomenon, as shown in Fig. 2b. Different from the bistable resistive switching phenomenon, the response current of our device gradually increases during application of five consecutive positive sweeps and decreases followed five consecutive negative sweeps, indicating tunable conductance characteristics which is analogue to the modulation of biological synapses.”

Figure R5. The current with respect to DC voltage sweeps, the device conductance gradually increase (decrease) under positive (negative) sweeps of 0 to 0.5 V (0 to -0.5 V).

(4.4) L293: you don't demonstrate the exact implementation of BCM.

We clarified the implementation of BCM learning as below: the implementation of BCM learning rule can be achieved based on a rational triplet-STDP scheme by elaborate designing the fire time of post-synaptic pulses. For instance, two post-synaptic pulses evenly distribute on the two sides of pre-synaptic pulse and the post-synaptic firing rate can be calculated as the reciprocal value of interval time of two post-synaptic pulses. The transition of memristor conductance states is identified as the result of BCM learning rule with a designed post-synaptic firing rate.

Following discussion was added in the revised manuscript:

“The implementation of BCM learning rule can be achieved based on a rational triplet-STDP scheme by elaborate designing the fire time of post-synaptic pulses. For instance, two post-synaptic pulses evenly distribute on the two sides of pre-synaptic pulse and the post-synaptic firing rate can be calculated as the reciprocal value of interval time of two post-synaptic pulses. The transition of memristor conductance states is identified as the result of BCM learning rule with a designed post-synaptic firing rate.”

(4.5) Fig 5h: why mean frequency start at 10 Hz why figure d and g start at 48 Hz?

The difference of the frequency start-values is because the input patterns are different in Fig. 5d, 5g and 5j. Fig. 5d demonstrates the formation of receptive field in normal binocular contour vision, the orientation inputs of left and right eyes are identical in every epoch. The input rates of grids on the orientation bar and background grids are 30 Hz and 14 Hz, respectively. While Fig. 5g demonstrates the formation of receptive field in monocular deprivation condition, the right eye is deprived, which is input with image containing noise pixels (4 to 6 Hz) and left eye is normal with input image of orientational pattern (30 Hz for orientation bar and 14 Hz for background noise). The high input rate can trigger about 48 Hz firing response. While in the Fig. 5j, the left and right eyes are both deprived with the input image of noisy optical pulse (4 to 6 Hz), the low input rate can only trigger about 10 Hz firing response.

(4.6) Fig 4f: how could you get a 400% change of conductance? All previous experiments are showing a change of 40 %.

In Fig. 4f, the vertical coordinate is not the percentage change of conductance, it is the conductance modulation as the function of time.

(4.7) Not clear what is rho_y: sum of all rho_x or is it a sequential experiment on each pixel? Is it Poisson distributed pulses? How do you manage synchronization issue if not?

ρ_y is the sum of all ρ_x as $\rho_y = G\rho_x = \sum_{i=1}^9 G_i\rho_x^i$, where the $G = [G_1, G_2 \dots G_i]$ and $\rho_x = [\rho_x^1, \rho_x^2 \dots \rho_x^i]$ are weights of synaptic array and input rate of input pattern, respectively.

Hence, the ρ_y is the product of G and ρ_x . All of the ρ_x and ρ_y are Poisson distributed.

During the simulation, a momentary break off scheme is used to solve synchronization issue: the inputs of pre-synapse are uninterrupted and received by post-synapse in the real-time.

When the post-synapse fires, the reception-process will momentary breaks off and resumes after fires.

Following discussion was added in the revised manuscript:

“During the learning process, the input voltage pulse is generated by solar cell when optical pulse arrives with a rate of ρ_x (Poisson distribution).”

“In this framework, simplified the constants, the numerical value of postsynaptic firing rate can be redefined as $\rho_y = G\rho_x$ with a Poisson distribution, too.”

“The post-cortical neuron responds to the input singles following a linear function with a firing

rate of $\rho_y = G\rho_x = \sum_{i=1}^9 G_i\rho_x^i$, where $G = [G_1, G_2 \cdots G_i]$ and $\rho_x = [\rho_x^1, \rho_x^2 \cdots \rho_x^i]$ (i is

the index of the synapse) are the synapse weight and input rate, respectively.”

Reviewer #2 (Remarks to the Author):

In this paper, the authors reported the bioinspired striate cortex with binocular and orientation-selective receptive field with the crossbar array of self-powered memristors. Each cross-point contained a perovskite solar cell directly stacking on a perovskite memristor. Furthermore, the plasticity of the flexible crossbar array of self-powered memristor was modulated with a generalized BCM learning rule for optical-encoded pattern recognition. This manuscript is well organized and contains technical advancements in this field. However, the experimental results in this manuscript are not enough to support the authors' claim. Authors need to improve the quality of the manuscript.

We are glad that the reviewer finds this work very interesting. We would like to thank the reviewer for taking the time to read the manuscript and provide insightful feedback that has helped us greatly in improving the quality of the manuscript. We have provided point-by-point answers to the comments and concerns raised by the reviewer. We have also highlighted the corresponding changes made to the manuscript.

- 1. The visual information is supplied to the device continuously. Therefore, the artificial striate cortex would be exposed to excessive inputs. To overcome these environments, the device should have high endurance characteristics. The authors need to provide additional data about the electrical endurance characteristics of the device and array.**

In order to evaluate the endurance characteristics of the device and array, we carried out the typical I - V characteristics of perovskite memristor under the voltage sweep from 0 to +1.5 V to 0 V to -1.5 V to 0 V with the compliance current (I_{cc}) of 30 mA, as shown in Fig. R6a. The memristor shows typical bipolar resistive switching behavior. During the positive scan (SET operation), The memristor initially exhibited a high resistance of 372Ω with the reading bias of 0.1 V. As the voltage increased to 0.94 V (set voltage), the memristor was abruptly switched to low resistance state (LRS) of 34Ω with the current reaching the I_{cc} . When the negative voltage applied on TE, the current suddenly decreased at -0.95 V (reset voltage) and device was

transited from LRS to HRS. The endurance characteristics is shown in the Fig. R6b. The resistance states show no degradation with acceptable variation in consecutive 1000 cycles. The HRS and LRS fluctuate in the range of [307Ω 401Ω] and [32Ω 56Ω], respectively with the switching window of ~8.1 which is comparable to the previous reported perovskite memristor. The Fig. R6c shows the retention characteristics of the memristor. Both HRS and LRS can be well preserved up to 10⁴ s, verifying that memristor could retain the conductance state in the long-term. In addition, 100 cycles characterizations have been implemented on 10 randomly selected memristors, the spatio-temporal distribution of HRS and LRS showed no obvious fluctuation which demonstrate the great C2C and D2D uniformity of the memristor. The variations of HRS and LRS, described as σ/μ (σ is the standard deviation and μ is the mean value), in the endurance, retention and mulit-device tests are calculated and provided in Table R3.

We have included the following discussion in the supporting information:

“We performed typical *I-V* characteristics of perovskite memristor under the voltage sweep from 0 to +1.5 V to 0 V to -1.5 V to 0 V with the compliance current (I_{cc}) of 30 mA, as shown in Figure S2a. The memristor shows typical bipolar resistive switching behavior. During the positive scan (SET operation), The memristor initially exhibited a high resistance of 372Ω with the reading bias of 0.1 V. As the voltage increased to 0.94 V (set voltage), the memristor was abruptly switched to low resistance state (LRS) of 34 Ω with the current reaching the I_{cc} . When the negative voltage applied on TE, the current suddenly decreased at -0.95 V (reset voltage) and device was transited from LRS to HRS. The endurance characteristics is shown in the Figure S2b. The resistance states show no degradation with acceptable variation in consecutive 1000 cycles. The HRS and LRS fluctuate in the range of [307Ω 401Ω] and [32 Ω 56Ω], respectively with the switching window of ~8.1 which is comparable to the previous reported perovskite memristor. The Figure S2c shows the retention characteristics of the memristor. Both HRS and LRS can be well preserved up to 10⁴ s, verifying that memristor could retain the conductance state in the long-term. In addition, 100 cycles characterizations have been implemented on 10 randomly selected memristors, the spatio-temporal distribution of HRS and LRS showed no obvious fluctuation which demonstrate the great C2C and D2D

uniformity of the memristor. The variations of HRS and LRS, described as σ/μ (σ is the standard deviation and μ is the mean value), in the endurance, retention and multi-device tests are calculated and provided in Table S1.”

Figure R6. a. Resistive switching of memristor with compliance current of 30mA. b. The endurance characteristic of the memristor. c. The retention characteristic of the memristor. d. the HRS and LRS distribution during consecutive 10 cycles from 10 randomly selected memristors.

Parameter	Endurance		Retention		Multi-Device	
	HRS	LRS	HRS	LRS	HRS	LRS
Mean Value	353.11 Ω	43.88 Ω	347.89 Ω	37.99 Ω	344.77 Ω	36.71 Ω
Standard Deviation	15.28 Ω	4.05 Ω	2.11 Ω	0.56 Ω	12.23 Ω	5.83 Ω
Variation	0.043	0.093	0.006	0.015	0.035	0.159

Table R3. The mean values, standard deviations and variations of HRS and LRS in the

endurance, retention and multi-device tests.

- 2. According to the manuscript, the authors utilized the perovskite material (CsPbBr₂I) for the memristor. Therefore, the memristor would have light reactivity when the optical inputs were applied to the device for the operation. The authors need to confirm that the self-powered devices were irrelevant to the light reactivity of the memristor. Also, the authors should provide additional explanations about the effect of the optical responses of the memristor during the device/array operation.**

This is an excellent question raised by the reviewer. The memristor is directly stacked onto the solar cell to form self-powered striate cortex which is then attached to hemispherical shaped substrate for the next-stage characterization. Therefore, the self-powered memristor is inverted on the hemispherical shaped substrate with solar cell on the top of memristor to receive the optical stimuli for pattern recognition (as illustrated in Fig. 4c). Sequentially, the solar cell would receive the optical illumination before memristor. By performing the UV-visible absorption spectrum of solar cell with the structure of ITO/SnO₂/CsFAPbI₃/Spiro/Au/ITO/Au, we confirm that the maximum transmittance of the solar cell is only 0.12% at the 322 nm, implying that almost negligible light could get through the solar cell to reach and influence the memristor.

We have included the following discussion in the supporting information:

“The memristor is directly stacked onto the solar cell to form self-powered striate cortex which is then attached to hemispherical shaped substrate for the next-stage characterization. Therefore, the self-powered memristor is inverted on the hemispherical shaped substrate with solar cell on the top of memristor to receive the optical stimuli for pattern recognition (as illustrated in Fig. 4c). Sequentially, the solar cell would receive the optical illumination before memristor. By performing the UV-visible absorption spectrum of solar cell with the structure of ITO/SnO₂/CsFAPbI₃/Spiro/Au/ITO/Au, we confirm that the maximum transmittance of the solar cell is only 0.12% at the 322 nm, implying that almost negligible light could get through the solar cell to reach and influence the memristor.”

Figure R7. The transmittance spectra of solar cell with structure of ITO/SnO₂/CsFAPbI₃/Spiro/Au/ITO/Au under 1-Sun illumination.

3. In Fig. 4, the size of the flexible array seemed to be large. What is the dimension of the array? Also, the presented array may need a large surface area to receive sufficient visual information from the environment whereas a small device size would be required to achieve the high-integration array for the recognition performance. Is there any plan to overcome these conflicting characteristics?

The dimension of the integrated artificial striate cortex array is 3 cm × 3 cm with the electrode width of 1000 μm. Each cross-point is an artificial striate cortex cell with the structure of ITO/CsPbBr₂I/P3HT/Au/ITO/Au/Spiro/CsFAPbI₃/SnO₂/ITO. As pointed by the reviewer, for receiving sufficient visual information from the environment, the array size is limited by the large size of sub-solar cell which hinders their integration in high density array. To overcome these conflicting characteristics, both developing small-sized perovskite solar cells/memristor with high performance and employing plano-convex lens to enlarge the Field-of-View (FoV) and to guide the focused light to the device are urgently required. 1) For scaling down the perovskite solar cell/memristor with satisfactory device performance, optimizing precursor solutions (compound engineering) and deposition methods (one step or two step, Lewis acid-base adduct method, gas or vacuum-assisted drying) with in-depth understanding of the

nucleation and crystal growth kinetics of perovskite is required to obtain pinhole free and smooth perovskite films. In addition, utilizing novel approaches including slot-die coating, screen printing, and electroplating, are expected to scale down electrodes to the well-defined small dimension (Nat. Mater. 2014, 13, 897; Acc. Chem. Res. 2016, 49, 311; Nano Energy 2014, 10, 10; AIChE Journal 2015, 61, 1745; Sol. RRL. 2021, 5, 2100381). 2) Concentrator photovoltaic (CPV) systems employing plano-convex lens to collect direct sunlight could concentrate the energy onto high integration array of small-sized striate cortex (Opt. Express. 2010 18, 1122). According to the previous reports, by delicately design the system with the lens coupled with waveguide, solar concentrator system could achieve 90% optical efficiency at $300 \times$ concentration (Fig. R8). Above-mentioned strategies ensure scaling down of the size of our artificial striate cortex array for integration into high density array in which the solar cell with satisfactory V_{oc} , J_{sc} and PCE could generate sufficient energy to drive perovskite memristor with well-modulated conductance states.

Figure R8. **a.** Solvent engineering steps for preparing perovskite films (Nat. Mater. 2014, 13, 897). **b.** The Lewis acid (A)-base (B) reacts to form an adduct (A·B). Lewis bases are divided into oxygen donors (O-donors), sulfur donors (S-donors), and nitrogen donors (N-donors) (Acc. Chem. Res. 2016, 49, 311). **c.** Schematic of gas-assisted spin coating method. (Nano Energy 2014, 10, 10) **d.** Schematic diagram of slot-die coating (AIChE Journal 2015, 61, 1745). **e.**

Schematic drawing of two masking concepts to fabricate plated copper electrodes on single-junction PSCs using different masking layers of full-area ALD Al_2O_3 layer and full-area PVD metal stack with self-passivated aluminum on top (Sol. RRL. 2021, 5, 2100381). f. A slab waveguide homogenizes and transports sunlight from all apertures to a single cell (Opt. Express. 2010, 18, 1122).

4. According to the manuscripts, the authors demonstrated the flexible array, which was the hemisphere shape. The authors need to provide the characteristics of the device/array depending on the different bending states.

Thanks for the reviewer's valuable comment. To investigate the flexibility of the device, we performed the pulse experiments on the device with different bending angles.

We have included the following discussion in the supporting information:

“By application of continuous 100 optical pulses and 100 electrical pulses, the corresponding potentiation and depression phenomenon as the function of bending angles was provided in Figure S15. The device exhibits stable LTP and LTD performance with bending angles ranging from 85° to 153° , implying its high flexibility.”

Figure R9. Typical potentiation and depression with respect to the different curvatures of the artificial striate cortex. The optical images of the memristor with different bending angles are shown in the upper panel.

5. In Fig. 3d, the authors showed the current response as a function of different light intensities. Is it possible to simulate the device/array with the function of light intensity? Also, is it possible to recognize the stripe patterns through simulation even when the light intensity and color map are mixed?

Thanks for the reviewer's valuable comment. As advised by the reviewer, current response as function of light irradiance with different intensities were included in the original manuscript (Fig. R10a), indicating that the conductance of memristor can be modulated by different light intensities. In addition, the higher light intensity could induce larger photovoltaic effect of solar cell to drive memristor to higher conductance states, corresponding to spike amplitude-based synaptic plasticity (Fig. R10b).

It is possible to recognize the stripe patterns through optical simulation with mixed light intensity and color map since the perovskite solar cell exhibits broadband response ranging from visible to near-infrared light and photovoltaic conversion efficiency is highly dependent on the light intensity. Since the focus of this article is to introduce the novel self-powered striate cortex and its benefit to implement rate-based BCM learning rule for binocular orientation selectivity, frequency-encoded optical stimuli is adequate for this proof-of-concept demonstration. Stripe patterns simulation with mixed light intensity and color map is kind of beyond scope and will be investigated in our future studies.

We included following discussion in the supplementary information:

“As shown in Figure S12a, the current response increases more significantly with the increase of light intensity, indicating that the conductance of memristor can be modulated by different light intensities. In addition, the higher light intensity could induce larger photovoltaic effect of solar cell to drive memristor to higher conductance states, corresponding to spike amplitude-based synaptic plasticity (Figure S12b).”

Figure R10. a. The current response as the function of 100 mW/cm², 50 mW/cm², 10 mW/cm² and 1 mW/cm². **b.** The potentiation behavior as the function of 100 mW/cm², 50 mW/cm², 10 mW/cm² and 1 mW/cm², and all the pulse width is set to 5ms.

6. According to the manuscript, the authors claimed that they emulated a binocular orientation selectivity. The main function of binocular vision in a bio-system is to determine the distance to the object through the phase difference between two eyes. On the other hand, binocular characteristics are not required to recognize the striped pattern. The simulated result of Fig. 5 was seemed to increase of recognition rate as the number of arrays increased rather than the effect of the binocular orientation (The higher recognition performance would be obtained from the number of devices or arrays other than effect of the binocular orientation.). Therefore, it seems difficult to express these characteristics as binocular functions.

We thank the referee for this constructive question.

In the biological visual system, both left and the right eyes receive the outside information and output signals to the striate cortex, V1 for first converge. Neurons in adult striate cortex are binocular with a strong selectivity for a particular orientation. For realizing normal binocular perceptions including stereopsis and depth, binocularly matched orientation selectivity between the two eyes is highly demanded. Binocular neurons in the striate cortex should match their orientation by tuning through the two eyes to perceive coherently.

In order to clarify the influence of array size on the orientation selectivity, we built five arrays with different sizes (9 by 9, 11 by 11, 13 by 13, 15 by 15 and 17 by 17) for orientation selectivity

simulation, and all of them can be trained successfully that one orientation can win the network and keep the winner position in the following training epochs. In addition, the orientation selectivity can be evaluated by:

$$\text{Orientation Selectivity} = 1 - \frac{\text{mean}(\rho_y^0, \rho_y^{45}, \rho_y^{90}, \rho_y^{135}) - \min(\rho_y^0, \rho_y^{45}, \rho_y^{90}, \rho_y^{135})}{\max(\rho_y^0, \rho_y^{45}, \rho_y^{90}, \rho_y^{135}) - \min(\rho_y^0, \rho_y^{45}, \rho_y^{90}, \rho_y^{135})}$$

where the $\rho_y^0, \rho_y^{45}, \rho_y^{90}$ and ρ_y^{135} are the response frequency for the orientation of $0^\circ, 45^\circ, 90^\circ$ and 135° , respectively. The simulation results were summarized in Table R4 and Fig. R11. It is clear that the orientation selectivity cannot be improved by increasing array size. In our work, the orientation bar is kind of small-sample pattern with limited number of pixels and pattern features, a 9 by 9 array can effectively perform feature extraction and classification during training and test processes for orientation selectivity task. Larger array size may induce the high energy consumption in this scenario.

We included following discussion in the supplementary information:

“In order to clarify the influence of array size on the orientation selectivity, we built five arrays with different sizes (9 by 9, 11 by 11, 13 by 13, 15 by 15 and 17 by 17) for orientation selectivity simulation, and all of them can be trained successfully that one orientation can win the network and keep the winner position in the following training epochs. In addition, the orientation selectivity can be evaluated by:

$$\text{Orientation Selectivity} = 1 - \frac{\text{mean}(\rho_y^0, \rho_y^{45}, \rho_y^{90}, \rho_y^{135}) - \min(\rho_y^0, \rho_y^{45}, \rho_y^{90}, \rho_y^{135})}{\max(\rho_y^0, \rho_y^{45}, \rho_y^{90}, \rho_y^{135}) - \min(\rho_y^0, \rho_y^{45}, \rho_y^{90}, \rho_y^{135})}$$

where the $\rho_y^0, \rho_y^{45}, \rho_y^{90}$ and ρ_y^{135} are the response frequency for the orientation of $0^\circ, 45^\circ, 90^\circ$ and 135° , respectively. The simulation results were summarized in Table S7 and Figure S20. It is clear that the orientation selectivity cannot be improved by increasing array size. In our work, the orientation bar is kind of small-sample pattern with limited number of pixels and pattern features, a 9 by 9 array can effectively perform feature extraction and classification during training and test processes for orientation selectivity task. Larger array size may induce the high energy consumption in this scenario. “

Figure R11. **a.** The final color maps of 9×9 cortical synapse arrays in normally rearing condition of orientation selectivity simulation. **b.** The evolution of synaptic weights as a function of simulation time. **c.** The evolution of postsynaptic firing frequencies and network threshold as a function of simulation time. **d-o.** The simulation results of 11×11 , 13×13 , 15×15 and 17×17 cortical synapse arrays.

Array Size	9×9	11×11	13×13	15×15	17×17
------------	--------------	----------------	----------------	----------------	----------------

Orientation Selectivity (%)	74.99	75.00	74.99	75.00	74.87
-------	-------	-------	-------	-------

Table R4. The orientation selectivity based on different array sizes.

7. According to the manuscript, the flexible array was presented in the form of a hemisphere. However, in such a structure, distortion could exist when recognizing straight objects. Did this structure and distortion affect the recognition of striped patterns? Also, is this reflected in the simulation? The authors need to provide additional data and explanations.

We thank the reviewer for this constructive question. The structure distortion between hemispherical artificial striate cortex and recognized straight objects will not affect the recognition process and performance, since the striate cortex shows uniform response to optical stimulation even from straight objects. We performed 50 pulse measurements from randomly selected 10 devices on the bended array (Fig. R12-13). The potentiation and depression process are triggered by consecutively 100 optical pulses of 100mW/cm²/5ms and consecutively 100 electrical pulses of 0.5V/5ms, respectively, all the optical pulses are penetrated from the top direction of hemispherical artificial striate cortex. It is clearly that the striate cortex can be programmed to multi states, and the variabilities of G_{\max} and G_{\min} , described as σ/μ (σ is the standard deviation and μ is the mean value), are calculated to be 0.018 and 0.164, respectively. In addition, the nonlinearity variabilities of LTP and LTD are only 0.431 and 0.046, indicating a great symmetricity of states modulation (Table R5).

Thanks to the great response uniformity, the striped patterns can be well recognized by our artificial striate cortex even the exist of structure distortion. In addition, we can focus the effort to the binocular interaction study in the simulation work without having to worry about the cortex response performance.

We included following discussion in the supplementary information:

“In order to investigate the uniformity of the artificial striate cortex, we performed 50 pulse experiments on 10 randomly selected devices on the bended array. The potentiation and depression process are triggered by consecutively 100 optical pulses with intensity of 100 mW/cm² and duration of 5 ms and consecutively 100 electrical pulses with amplitude of 0.5 V and duration of 5 ms, respectively, all the optical pulses are penetrated from the top direction

of hemispherical artificial striate cortex. It is clearly that the device can be programmed to multi states, and the variabilities of G_{\max} and G_{\min} , described as σ/μ (σ is the standard deviation and μ is the mean value), are calculated to be 0.018 and 0.164, respectively. In addition, the nonlinearity variabilities of LTP and LTD are only 0.431 and 0.046, indicating a great symmetricity of states modulation.”

Figure R12. The potentiation and depression characteristics of the randomly selected device 1-5.

Figure R13. The potentiation and depression characteristics of the randomly selected device 6-10.

Parameter	G_{\max}	G_{\min}	α_p	β_p	α_d	β_d
Mean Value	439.77 μS	47.47 μS	10.72 μS	1.59	46.72 μS	3.88

Standard Deviation	7.97 μ S	7.78 μ S	5.65 μ S	0.69	6.43 μ S	0.18
Variation	0.018	0.164	0.527	0.434	0.138	0.046

Table R5. The mean values, standard deviations and variations of HRS, LRS and nonlinearity in the pulse experiments.

Reviewer #3 (Remarks to the Author):

In this work, the authors fabricate a self-powered memristor array powered by a directly integrated solar-cell which makes the memristor can be modulated by the external light pulses just like photosensory neurons and synapses in the human retina. The proposed memristor also has a second-order characteristics, making it possible to control the memristor with non-overlapping pulses in STDP or SRDP learning rule, which enables bio-plausible memristor based neuromorphic system. Based on the self-powered and second-order effect of the memristor-solar cell integrated array, the authors demonstrated the orientation selectivity in binocular condition with BCM learning rule.

The novelties of this manuscript are 1) BCM learning rule from self-powered memristor array and 2) binocular orientation selectivity using the memristor based system. However, the first novelty is already discussed from the authors' lab (ref. 40), and BCM learning rule using memristors has been proved by others work (ex. Ref. 50). The second novelty is valid while monocular orientation selectivity has been discussed in others work (ex. 50) In other words, previous studies such as reference 40 and 50 in the paper already demonstrated the self-power memristor by directly integrating a solar cell and a memristor, and the monocular orientation selectivity with BCM learning rule based on the second-order memristors, but the orientation selectivity experiment based on the self-powered memristor has an advantage in that the optical signal directly affects the synapse similar to the photosensory neuron in the human's visual system.

The integration of the memristor device and the solar cell is interesting, and the second-order effect of the memristor is highly desirable for a bio-plausible applications, however, there are several points that the authors should provide more detailed explanations and they should clarify the significance of the work (binocular orientation selectivity) compared to the previous works, considering that the three of advantages in the paper i) binocular orientation selectivity, ii) BCM learning rule in second order memristor, and iii) self-powered memristor by integrating the solar cell and the memristor device have been demonstrated already. Please see below for more detailed comments.

We are glad that the reviewer finds this work interesting. We appreciate your insightful comments on our research. We have revised the manuscript according to your suggestions and believe that these revisions have improved this work.

First, the differences and novelty of our work are highlighted as below:

(1) Binocular orientation selectivity: In the biological visual system, both left and the right eyes receive the outside information and output signals to the striate cortex, V1 for first converge. Neurons in adult striate cortex are binocular with a strong selectivity for a particular orientation. For realizing normal binocular perceptions including stereopsis and depth, binocularly matched orientation selectivity between the two eyes is highly demanded. Binocular neurons in the striate cortex should match their orientation by tuning through the two eyes to perceive coherently. As pointed out by the reviewer, reference 50 reported emulation of monocular orientation selectivity with BCM learning rule based on the second-order memristors (only functioning as synapse) without considering the correlated neuronal activity between the eye specific inputs to individual cortical neurons. In our work, the hardware implementation of artificial striate cortical neuron with binocular orientation selectivity based on self-powered memristor (combining the functions of both sensory neuron, cortical synapse and striate cortical neuron) has been first reported.

(2) Solution-processed monolithic all-perovskite system for self-powered artificial striate cortex: our previous work (reference 40) just demonstrated the self-powered artificial retina system by wire-bonding the silicon solar cell with perovskite memristor where the solar cell and memristor are separated. Specific interconnections between solar cell and memristor with inherent heterotypic materials restricts the simplifying process complexity and reducing packing density. However, in this work, the crossbar array of artificial striate cortex is the solution-processed monolithic all-perovskite system where each cross-point contains one CsFAPbI₃ perovskite solar cell directly stacking on the CsPbBr₂I perovskite memristor. Solution-processable metal halide perovskites are attractive for high-efficiency, low-cost solar cells owing to their tunable bandgaps, high optical absorption coefficient, simple processing and high defect tolerance. In addition, benefiting from excellent ionic dynamic properties, perovskites are suitable for developing memristors with adjustable conductance properties and low power consumption. It remains challenging to fabricate solution-processed monolithic all-

perovskite system for striate cortex since solution processing of a perovskite film often induces damage to the underlying as-formed perovskite layers. It is also challenging to optimize the charge transport interface to ensure that solar cell could supply sufficient power to drive memristor. By delicately compound engineering and structural design, we first realize solution-processed monolithic all-perovskite system to implement the hardware-based striate cortex with binocular and orientation selective receptive field.

In summary, our work first reported hardware implemented striate cortex based on solution-processed monolithic all-perovskite system with binocular and orientation selectivity, as far as we understand. The bio-inspired striate cortex is highly compatible with high-density and low power consumption machine vision owing to its crossbar paradigm and homotypic materials system.

- 1. The authors claim second order phenomena because the device has two time constants: one is for long-term steady state and the other is for short-term volatile state. However, the authors focused on the short-term volatile state and did not explain the behavior of long-term state. The main part of this manuscript is “memristor” device. The authors should provide “basic information” about memristors such as set voltage, reset voltage, endurance, retention, on/off ratio and spatio-temporal distributions. Without that information, the device cannot be verified. what is the condition of the short-term to long-term transition of the device, and how long the device retain the conductance state in the long-term and the short-term mode?**

We thank the referee for this constructive suggestion. According to reviewer’s suggestion, we performed typical I - V characteristics of perovskite memristor under the voltage sweep from 0 to +1.5 V to 0 V to -1.5 V to 0 V with the compliance current (I_{cc}) of 30 mA, as shown in Fig. R14a. The memristor shows typical bipolar resistive switching behavior. During the positive scan (SET operation), The memristor initially exhibited a high resistance of 372Ω with the reading bias of 0.1 V. As the voltage increased to 0.94 V (set voltage), the memristor was abruptly switched to low resistance state (LRS) of 34Ω with the current reaching the I_{cc} . When the negative voltage applied on TE, the current suddenly decreased at -0.95 V (reset voltage) and device was transited from LRS to HRS. The endurance characteristics is shown in the Fig.

R14b. The resistance states show no degradation with acceptable variation in consecutive 1000 cycles. The HRS and LRS fluctuate in the range of [307 Ω 401 Ω] and [32 Ω 56 Ω], respectively with the switching window of ~ 8.1 which is comparable to the previous reported perovskite memristor. The Fig. R14c shows the retention characteristics of the memristor. Both HRS and LRS can be well preserved up to 10^4 s, verifying that memristor could retain the conductance state in the long-term. In addition, 100 cycles characterizations have been implemented on 10 randomly selected memristors, the spatio-temporal distribution of HRS and LRS showed no obvious fluctuation which demonstrate the great C2C and D2D uniformity of the memristor. The variations of HRS and LRS, described as σ/μ (σ is the standard deviation and μ is the mean value), in the endurance, retention and multi-device tests are calculated and provided in Table R6.

To further clarify the co-existence of short-term and long-term dynamics of memristor, we performed a series of pulse experiments on the memristor with the pulse number ranging from one to five. The conductance of memristor decays to original state of $110\ \mu\text{S}$ after application of one single pulse while after application of five pulses, the conductance can be stabilized at $150\ \mu\text{S}$ for 10^4 s, indicating the transition from short-term dynamics to long-term dynamics (Fig. R15). In addition, we re-fit PPF curve of memristor to obtain the STP time constants (τ_1) of 0.28 ms and LTP time constants (τ_2) of 10.86 ms (Table R7). Apparently, τ_2 is about one order of magnitude larger than τ_1 , which agrees well with the measured data in biological synapses (\sim ms order) and other artificial synapse works (Adv. Electron. Mater. 2019, 5, 1900287; Nat. Electron. 2021, 4, 348; Nat. Mater. 2016, 16, 101). The τ_1 and τ_2 are the time constants of relaxation processes, the states can be stabilized for 10^4 s indicating the non-volatile characteristic of our memristor.

We have included the following discussion in the supporting information or revised manuscript: “We performed typical I - V characteristics of perovskite memristor under the voltage sweep from 0 to +1.5 V to 0 V to -1.5 V to 0 V with the compliance current (I_{cc}) of 30 mA, as shown in Figure S2a. The memristor shows typical bipolar resistive switching behavior. During the positive scan (SET operation), The memristor initially exhibited a high resistance of $372\ \Omega$ with the reading bias of 0.1 V. As the voltage increased to 0.94 V (set voltage), the memristor

was abruptly switched to low resistance state (LRS) of 34Ω with the current reaching the I_{cc} . When the negative voltage applied on TE, the current suddenly decreased at -0.95 V (reset voltage) and device was transited from LRS to HRS. The endurance characteristics is shown in the Figure S2b. The resistance states show no degradation with acceptable variation in consecutive 1000 cycles. The HRS and LRS fluctuate in the range of [307Ω 401Ω] and [32Ω 56Ω], respectively with the switching window of ~ 8.1 which is comparable to the previous reported perovskite memristor. The Figure S2c shows the retention characteristics of the memristor. Both HRS and LRS can be well preserved up to 10^4 s, verifying that memristor could retain the conductance state in the long-term. In addition, 100 cycles characterizations have been implemented on 10 randomly selected memristors, the spatio-temporal distribution of HRS and LRS showed no obvious fluctuation which demonstrate the great C2C and D2D uniformity of the memristor. The variations of HRS and LRS, described as σ/μ (σ is the standard deviation and μ is the mean value), in the endurance, retention and multi-device tests are calculated and provided in Table S1.”

“By fitting the PPF curve, the τ_1 of STP constant and τ_2 of LTP constant were estimated as 0.28 ms and 10.86 ms, respectively. The τ_2 is one order larger than τ_1 which is comparable to the biological synapse (\sim ms order). “

Figure R14. **a.** Resistive switching of memristor with compliance current of 30mA. **b.** The endurance characteristic of the memristor. **c.** The retention characteristic of the memristor. **d.** the HRS and LRS distribution during consecutive 10 cycles from 10 randomly selected memristors.

Parameter	Endurance		Retention		Multi-Device	
	HRS	LRS	HRS	LRS	HRS	LRS
Mean Value	353.11 Ω	43.88 Ω	347.89 Ω	37.99 Ω	344.77 Ω	36.71 Ω
Standard Deviation	15.28 Ω	4.05 Ω	2.11 Ω	0.56 Ω	12.23 Ω	5.83 Ω
Variation	0.043	0.093	0.006	0.015	0.035	0.159

Table R6. The mean values, standard deviations and variations of HRS and LRS in the endurance, retention and multi-device tests.

Figure R15. **a.** The biological memory model proposed by Atkinson and Shiffrin. **b.** The memristor current responses to pulse train stimulations with different pulse numbers.

Parameter	c_1	τ_1	c_2	τ_2
Value	0.7547	0.28 ms	0.4663	10.86 ms

Table R7. The fitting results of PPF.

2. Usually, perovskite based memristors have very “short life time”, which means the device characteristics are degraded in a few days. The authors should claim the life time of the device and why they chose perovskite materials for this application.

In order to evaluate the life time of our memristor, we carry out the potentiation and depression measurements with respect to the time eclipse (Fig. R16). The memristors were applied with 100 consecutively positive pulses with amplitude of 0.5 V and duration of 5 ms followed by 100 consecutively negative pulses with amplitude of -0.5 V and duration of 5 ms. Thanks to the encapsulation by glass and edge sealed by optical adhesive, the LTP and LTD characteristics of memristor shows no obvious degradation after 10, 20 and 30 days.

The perovskites are prospective for imitating the ion dynamics on the membranes of biological neurons because of its intrinsic ion migration. The external electrical/optical-triggered ion drift and diffusion, which is analogue with the influx and extrusion of Ca^{2+} through the synaptic cell, are identified as the switching mechanism of perovskite memristor. Owing to the lower drift and diffusion barriers ($\approx 0.16\text{eV}$), (Adv. Theory Simul. 2018, 1, 1700035) the perovskite memristors possess low power consumption ($\approx 10\text{ fJ}$) (Nano Energy 2018, 48, 575), multi-level data modulation (>1000 states) (Nano Energy 2020, 74, 104828), large switching window (\approx

10⁷) (Adv. Mater. 2017, 29, 1700527; Front. Neurosci. 2021, 15, 661856) and low operation voltage ($\approx 0.13\text{V}$) (Adv. Mater. 2016, 28, 6562). In addition, the low-temperature solution-processed perovskite are compatible with low cost and flexible integrated circuits.

The reviewer's concern about the stability of perovskite memristor is indeed. Compared with traditional memristor based on stable metal oxide, the lifetime of perovskite memristor is relatively shorter. Considering the excellent ion migration behavior in perovskite which is suitable for emulation of synapse with STP and LTP, the lifetime of perovskite memristor can be effectively extended by encapsulation approaches such as stack blanket encapsulation scheme and edge encapsulation scheme etc (Science 2020, 368, eaba2412).

We included following discussion in the supplementary information:

“In order to evaluate the life time of our memristor, we carry out the potentiation and depression measurements with respect to the time eclipse (Figure S4). The memristors were applied with 100 consecutively positive pulses with amplitude of 0.5 V and duration of 5 ms followed by 100 consecutively negative pulses with amplitude of -0.5 V and duration of 5 ms. Thanks to the encapsulation by glass and edge sealed by optical adhesive, the LTP and LTD characteristics of memristor shows no obvious degradation after 10, 20, 30 days.”

Figure R16. The obtained conductance potentiation and depression after **a.** 10 days **b.** 20 days **c.** 30 days.

3. Perovskite patterning: The authors did not show the real image of the array, even the single cell image. It is well known that it is very hard to form crossbar structure with perovskite due to fabrication issue. The authors should provide the real image of the

array and stand-alone device. The information about device size needs to be provided, too.

Thanks for reviewer's constructive question. Note that the real image of the array and the cross-sectional SEM image of the single self-power memristor with perovskite solar cell directly stacked on the perovskite memristor was included in the Fig. 3a and Fig. 4b. The continuous and smooth surface of each layer can be observed, verifying that our design of solution-processed monolithic all-perovskite system allows the facile integration of memristor and solar cell without damaging of each layer.

Reviewer's concern regarding compatibility of photovoltaic cells and memristors in term of fabrication issue is indeed. It remains challenging to fabricate solution-processed monolithic all-perovskite system for striate cortex since solution processing of a perovskite film often induces damage to the underlying as-formed perovskite layers. It is also challenging to optimize the charge transport interface to ensure that solar cell could supply sufficient power to drive memristor. By delicately compound engineering and structural design, we first realize solution-processed monolithic all-perovskite system to implement the hardware-based striate cortex with binocular and orientation selective receptive field.

The details about the design of the memristor and photovoltaic cell are as below: we first fabricated solar cell by sandwiching a CsFAPbI₃ perovskite active layer between two selective charge transport layers and electrodes where SnO₂ is electron transport layer for negative charge extraction, spiro-OMeTAD film functions as hole transports layer for positive charge extraction, ITO is anode and Au electrode functions as cathode. Subsequently, we directly integrated memristor onto the as-fabricated solar cell by the following procedures: the 600 nm ITO layer were deposited onto the Au cathode of solar cell by magnetron sputtering which are compact enough to protect the as-formed sub-solar cell from damage during solution processing of perovskite memristor. Then, the Au top electrodes were deposited on ITO by thermal evaporation through a metal shadow mask. Ultrathin P3HT film was spin-coated onto Au layer to function as reservoir layer to accept the migrated halide ions for continuous modulation of conductance. The CsPbBr₂I was spin-coated on the P3HT film to act as switching layer. Finally, the ITO BE was deposited by vacuum evaporation.

Following discussion was added in the revised manuscript:

“We first fabricated solar cell by sandwiching a perovskite active layer between two selective charge transport layers and electrodes where SnO_2 is electron transport layer for negative charge extraction, spiro-OMeTAD film functions as hole transports layer for positive charge extraction, ITO is anode and Au electrode functions as cathode. The colloidal solution of SnO_2 (3.75 wt%) was spin-coated on the ITO substrates at 3000 rpm for 30 s and then the SnO_2 -coated ITO substrates were annealed at 150°C for 30 min. It's worth noting that the diluted SnO_2 colloidal solution was filter by the $0.45\ \mu\text{m}$ PVDF filter before use. Then, the CsFAPbI_3 perovskite film was prepared by one-step spin coating method and followed an annealing process at 150°C for 30 min. The filtered Spiro-OMeTAD solution by the $0.45\ \mu\text{m}$ PTFE filter was spin coated onto the perovskite layer at 3,000 r.p.m. for 30 s. The Au cathode were deposited by thermal evaporation through a metal shadow mask. Subsequently, we directly integrated memristor onto the as-fabricated solar cell by the following procedures: the 600 nm ITO layer were deposited onto the Au cathode of solar cell by magnetron sputtering which are compact enough to protect the as-formed sub-solar cell from damage during solution processing of perovskite memristor. Then, the Au top electrodes were deposited on ITO by thermal evaporation through a metal shadow mask. Ultrathin P3HT film was spin-coated onto Au layer to function as reservoir layer to accept the migrated halide ions for continuous modulation of conductance. The CsPbBr_2I was spin-coated on the P3HT film to act as switching layer. Finally, the ITO BE was deposited by vacuum evaporation.”

Figure R17. The illustration of artificial striate cortex array and the cross-sectional SEM image of the single cell.

4. Sneak path problem: How did the authors avoid the sneak path problem in memristor array? The I-V curve from figure 2b does not show any rectification.

Thanks for the reviewer’s valuable comment. The sneak path problem was avoided by employing programming-inhibit operation in our work. As shown in Fig. R18, the bit line of the selected memristor was applied a programming pulse (0.5V/5ms) and the word line was set to 0 V. In the meantime, all the bit lines and word lines of unselected memristors were applied an inhibit pulses (0.25V/10ms) to prevent the programming disturbance. The application of inhibiting pulses reduces the potential drop between the word line and bit line of unselected memristor which can protect the unselected memristor from state changes.

We included following discussion in the supplementary information:

“The sneak path problem was avoided by employing programming-inhibit operation in our work. As shown in Figure S18, the bit line of the selected memristor was applied a programming pulse (0.5V/5ms) and the word line was set to 0 V. In the meantime, all the bit lines and word lines of unselected memristors were applied an inhibit pulses (0.25V/10ms) to prevent the programming disturbance. The application of inhibiting pulses reduces the potential drop between the word line and bit line of unselected memristor which can protect the unselected memristor from state changes.”

Figure R18. Weight updating method.

5. The authors should provide information about solar cell and memristor devices such as solar cell output voltage, memristor input voltage range, etc.

We would like to thank reviewer for this valuable suggestion. The detailed parameters about

solar cell (Table R8) were included in the revised supporting information. In order to clarify the memristor input voltage range, we carried out the consecutive pulse tests with different pulse amplitudes and fixed pulse width, as shown in Fig. R19. It can be seen that the memristor can be modulated by pulse amplitude of $\pm 0.5V$, $\pm 0.4V$ and $\pm 0.3V$, while the pulse of $\pm 0.2V$ and $\pm 0.1V$ cannot trigger the memristor.

We included following discussion in the supplementary information:

“In order to clarify the memristor input voltage range, we carried out the consecutive pulse tests with different pulse amplitudes and fixed pulse width, as shown in Figure S5. It can be seen that the memristor can be modulated by pulse amplitude of $\pm 0.5V$, $\pm 0.4V$ and $0.3V$, while the pulse of $\pm 0.2V$ and $\pm 0.1V$ cannot trigger the memristor.”

Parameter	Area	I_{sc}	V_{oc}	P_{max}	I_{max}	V_{max}	Efficiency	Fill Factor	J_{sc}	J_{max}
Value	0.02cm ²	0.47mA	1.1V	0.43mW	0.45mA	0.95V	21.6%	82.6%	23.7 mA/cm ²	22.7 mA/cm ²

Table R8. The details of the solar cell employed in this work.

Figure R19. LTP and LTD characteristics of memristor by applying various pulse voltages of $\pm 0.5V$, $\pm 0.4V$, $\pm 0.3V$, $\pm 0.2V$ and $\pm 0.1V$, and all the pulse width is set to 0.5ms.

6. No explanation about figure 1d in the manuscript.

Thanks for the reviewer's valuable suggestion. We included the explanation of Fig. 1d in the revised manuscript as below: Fig. 1d shows the schematic illustration of development of binocular, orientation preference receptive field in the human visual system driven by visual experience based on the BCM rate-based theory with typical feature of sliding threshold.

7. The authors should provide the uniformity value from pulse measurement. If the variation is large, the figures (for example figure 4d) cannot be valid.

Thanks for the reviewer's valuable suggestion. In order to investigate the uniformity of the artificial striate cortex, we performed 50 pulses experiments on 10 randomly selected devices (Fig. R20-21). In addition, by tuning synapse history state G_0 , we implemented BCM learning rules as a function of postsynaptic firing frequency 100 times to include the error bar in the experimental data (Fig. R22).

We included following discussion in the supplementary information and updated Fig. 4d:

“In order to investigate the uniformity of the artificial striate cortex, we performed 50 pulse experiments on 10 randomly selected devices. The potentiation and depression process are triggered by consecutively 100 optical pulses with intensity of $100\text{mW}/\text{cm}^2$ and duration of 5 ms and consecutively 100 electrical pulses with amplitude of 0.5V and duration of 5 ms, respectively, all the optical pulses are penetrated from the top direction of hemispherical artificial striate cortex. It is clearly that the device can be programmed to multi states, and the variabilities of G_{\max} and G_{\min} , described as σ/μ (σ is the standard deviation and μ is the mean value), are calculated to be 0.018 and 0.164, respectively. In addition, the nonlinearity variabilities of LTP and LTD are only 0.431 and 0.046, indicating a great symmetricity of states modulation.”

Figure R20. The potentiation and depression characteristics of the randomly selected device 1-5.

Figure R21. The potentiation and depression characteristics of the randomly selected device 6-10.

Parameter	G_{\max}	G_{\min}	α_p	β_p	α_d	β_d
Mean Value	439.77 μS	47.47 μS	10.72 μS	1.59	46.72 μS	3.88

Standard Deviation	7.97 μ S	7.78 μ S	5.65 μ S	0.69	6.43 μ S	0.18
Variation	0.018	0.164	0.527	0.434	0.138	0.046

Table R9. The mean values, standard deviations and variations of HRS, LRS and nonlinearity in the pulse experiments.

Figure R22. The measured (triangle) and fitting (line) results of BCM learning rule as a function of postsynaptic firing frequency based on the artificial striate cortex.

8. Please specify the figure number of supporting information in the manuscript. The authors just mentioned supporting information without any specific figure numbers.

Thanks for the reviewer's valuable suggestion. We have updated the figure number of Supporting Information in the revised manuscript.

9. What did the authors want to show from figure 3e and 3f? It is natural that the device has negative current with post-pulse and positive current with pre-pulse.

Thanks for the reviewer's valuable suggestion. Note that the Fig. 3d-3f have been moved to Supporting Information.

10. In 3 x 3 array test, how did the authors select specific cell for the light pattern? From

the system, the light emits from outside and then all devices might receive the same light.

We thank the referee for this technical question. For the implementation of light pattern recognition, the array of artificial striate cortex was divided to two groups based on different frequency ranges of optical input signal. One group consists of five cells whose input is high-frequency optical signal and another group contains four cells whose input is low-frequency optical signal. The cells in the same group triggered by the same optical stimuli can be operated synchronously which greatly simplified our experimental setup. The optical signal with different frequency was generated by combining the normal light source with rotating disc optical chopper.

We included following discussion in the Methods:

“For the implementation of light pattern recognition, the array of artificial striate cortex was divided to two groups based on different frequency ranges of optical input signal. One group consists of five cells whose input is high-frequency optical signal and another group contains four cells whose input is low-frequency optical signal. The cells in the same group triggered by the same optical stimuli can be operated synchronously which greatly simplified our experimental setup. The optical signal with different frequency was generated by combining the normal light source with rotating disc optical chopper.”

11. With the scheme in figure 4d, the sliding threshold effect is just a consequence of the previous history, and it is not controllable parameter. And this behavior is common in general volatile memristors. The authors should comment how to control this sliding threshold effect. For example, the homeostasis, which is for controlling the number of spikes, can be controlled by gate voltage in 3 terminal devices. Also, is there any improvement by this sliding threshold effect on your application? The authors should comment on this, too.

We thank the referee for this insightful question. In BCM learning rule, the sliding threshold is a dependent variable which is used to judge the potentiation or depression behavior under external excitation. It is associated with previous history of device that a high device conductance state is prone to trigger potentiation and vice versa. The sliding threshold can be

indirectly controlled by adjusting the device conductance: the left or right sliding is obtained by the decrease or increase of device conductance with application of consecutive pre- or post-spikes. This control method is also in consist with its definition that the threshold is a history dependent (the consecutive spike excitation over the period) variable.

The sliding characteristic is designed to solves the unstable issue of the system. For instance, in the fixed threshold system, setting the threshold too low causes synaptic strength to grow without bound while setting the threshold too high induce all synapses weaken to zero. By introducing the sliding characteristic, the threshold can be automatically adjusted by device history states, and the BCM learning rule overcome the limitations of the fixed threshold system and yield both selectivity and stability. In other words, the sliding threshold gives the dynamic adaptability of visual system.

12. The authors performed simulation by using 9 by 9 array. 9 by 9 array can be achieved experimentally. The authors should explain why larger array size was not utilized for the simulation. 5. Also the simulation seems that only one orientation was randomly learned. Are the other three orientation were learned with similar probability?

We thank the referee for this constructive question. We agree with the reviewer that the array size of 9 by 9 is not as large as that used in the previous simulation works. In order to clarify the influence of array size on the orientation selectivity, we built five arrays with different sizes (9 by 9, 11 by 11, 13 by 13, 15 by 15 and 17 by 17) for orientation selectivity simulation, and all of them can be trained successfully that one orientation can win the network and keep the winner position in the following training epochs. In addition, the orientation selectivity can be evaluated by:

$$\text{Orientation Selectivity} = 1 - \frac{\text{mean}(\rho_y^0, \rho_y^{45}, \rho_y^{90}, \rho_y^{135}) - \min(\rho_y^0, \rho_y^{45}, \rho_y^{90}, \rho_y^{135})}{\max(\rho_y^0, \rho_y^{45}, \rho_y^{90}, \rho_y^{135}) - \min(\rho_y^0, \rho_y^{45}, \rho_y^{90}, \rho_y^{135})}$$

where the $\rho_y^0, \rho_y^{45}, \rho_y^{90}$ and ρ_y^{135} are the response frequency for the orientation of $0^\circ, 45^\circ, 90^\circ$ and 135° , respectively. The simulation results were summarized in Table R10 and Fig. R23. It is clear that the orientation selectivity cannot be improved by increasing array size. In our work, the orientation bar is kind of small-sample pattern with limited number of pixels and pattern features, a 9 by 9 array can effectively perform feature extraction and classification

during training and test processes for orientation selectivity task. Larger array size may induce the high energy consumption in this scenario.

In addition, we performed orientation selectivity simulation for different orientation of 45°, 0° and 135°, as shown in Fig. R24. All of them displays similar results with the orientation of 90° in the manuscript. And 100 repeat simulations are performed to count the probability of different orientation selectivity, as summarized in Table R11. The statistical result shows that all the orientation can be learned with similar probability.

We included following discussion in the supplementary information:

“In order to clarify the influence of array size on the orientation selectivity, we built five arrays with different sizes (9 by 9, 11 by 11, 13 by 13, 15 by 15 and 17 by 17) for orientation selectivity simulation, and all of them can be trained successfully that one orientation can win the network and keep the winner position in the following training epochs. In addition, the orientation selectivity can be evaluated by:

$$\text{Orientation Selectivity} = 1 - \frac{\text{mean}(\rho_y^0, \rho_y^{45}, \rho_y^{90}, \rho_y^{135}) - \min(\rho_y^0, \rho_y^{45}, \rho_y^{90}, \rho_y^{135})}{\max(\rho_y^0, \rho_y^{45}, \rho_y^{90}, \rho_y^{135}) - \min(\rho_y^0, \rho_y^{45}, \rho_y^{90}, \rho_y^{135})}$$

where the ρ_y^0 , ρ_y^{45} , ρ_y^{90} and ρ_y^{135} are the response frequency for the orientation of 0°, 45°, 90° and 135°, respectively. The simulation results were summarized in Table S7 and Figure S20. It is clear that the orientation selectivity cannot be improved by increasing array size. In our work, the orientation bar is kind of small-sample pattern with limited number of pixels and pattern features, a 9 by 9 array can effectively perform feature extraction and classification during training and test processes for orientation selectivity task. Larger array size may induce the high energy consumption in this scenario.

In addition, we performed orientation selectivity simulation for different orientation of 45°, 0° and 135°, as shown in Figure S21. All of them displays similar results with the orientation of 90° in the manuscript. And 100 repeat simulations are performed to count the probability of different orientation selectivity, as summarized in Table S8. The statistical result shows that all the orientation can be learned with similar probability. ”

Figure R23. **a.** The final color map of 9×9 cortical synapse array in normally rearing condition of orientation selectivity simulation. **b.** The evolution of synaptic weights as a function of simulation time. **c.** The evolution of postsynaptic firing frequencies and network threshold as a function of simulation time. **d-o.** The simulation results of 11×11 , 13×13 , 15×15 and 17×17 cortical synapse arrays.

Array Size	9×9	11×11	13×13	15×15	17×17
Orientation Selectivity	74.99	75.00	74.99	75.00	74.87

(%)					
-----	--	--	--	--	--

Table R10. The orientation selectivity based on different array sizes.

Figure R24. **a.** The final color maps of 9×9 cortical synapse arrays with the 45° as the winner of orientation selectivity simulation, including two arrays for left and right eyes. **b.** The evolution of synaptic weights as the function of simulation time, where the upper and lower behalf are corresponding to left and right eyes. **c.** The evolution of postsynaptic firing frequencies and network threshold as a function of simulation time. It is clearly that only ρ_y^{45} is larger than firing threshold which means the vertical is the selected orientation. **d-i.** Similarly, the simulation results with the winner orientation of 0° and 135° .

Orientation	0°	45°	90°	135°
Time	23	26	26	25

Table R11. The winner time of four orientations in 100 simulations.

13. In line 302, the SNN is trained with winner-take-all method, which usually represents a biological phenomenon that the firstly firing neuron inhibits the other neurons in the same layer to effectively reduce the unnecessary firing event and prevent learning of wrong information. In the paper, it is written that the winner-take-all method is applied among four different firing rates in a same post-synaptic neuron. It is

confusing that how the winner-take-all method is applied to the given system, where there is only a single output neuron.

We apologize for the ambiguous statements. Indeed, the post-synaptic neuron synchronously fires based on the real-time pre-synaptic inputs, no matter which orientation is input. And the ‘winner-take-all’ in the manuscript means that the winner orientation can maintain its competitive advantage during the network training and inhibits other orientations shake its winner state.

We have revised the explanation in the manuscript as follows:

“During each simulation epoch, the four orientation patterns are stochastically input to the right and left eyes with equal probability, and the postsynaptic neuron synchronously responds based on the real-time presynaptic input with the firing rate of ρ_y . Then, the synaptic weight G and the threshold θ are modified following BCM learning rule in real-time. Finally, the orientation bar which can drives the highest postsynaptic firing rate will selected by the network, serving as the winner orientation and inhibiting other orientations shake its winner state.”

14. Because the self-power memristor characteristics, the demonstration of the sliding effect in BCM learning rule in memristors, and the orientation selectivity in the memristor based SNN are already studied, the strongest novelty in the paper is in the binocular orientation selectivity based on the self-power memristor, that can directly utilize the optical signal. However, it is not clear that what is the advantage of the binocular condition in the orientation selectivity. The authors claimed that the simulation results are well matched with experimental findings in line 326, but it would be better to emphasize the advantage of the demonstration in binocular condition.

Thanks for reviewer’s valuable comments. The novelty and advantage of the binocular condition in the orientation selectivity is highlighted as below: in the biological visual system, signals from the left and the right eyes first converge in the striate cortex, V1. Neurons in adult striate cortex are binocular with a strong preference for contours of a particular orientation. The newly born interocular neurons have different orientation preference which means the binocular

response is inconsistent for the same field of view (FoV). A matching process between two eyes is required to form normal binocular perception for depth and stereopsis. Binocular neurons in the striate cortex must match their orientation tuning through the two eyes in order for the animal to perceive coherently. In addition, the ocular dominance shift plasticity of orientation selectivity, which describes the competition effect between two eyes under different rearing condition is also based on binocular condition. As pointed out by the reviewer, emulation of monocular orientation selectivity with BCM learning rule has been reported. However, this work was based on the second-order memristors (only functioning as synapse) without considering the correlated neuronal activity between the eye specific inputs to individual cortical neurons. In our work, the hardware implementation of artificial striate cortical neuron with binocular orientation selectivity based on self-powered memristor (combining the functions of both sensory neuron, cortical synapse and striate cortical neuron) has been first reported.

We enriched the novelty description about binocular orientation selectivity in the revised manuscript as follows:

“In the biological visual system, signals from the left and the right eyes first converge in the striate cortex, V1. Neurons in adult striate cortex are binocular with a strong preference for contours of a particular orientation. The newly born interocular neurons have different orientation preference which means the binocular response is inconsistent for the same field of view (FoV). A matching process between two eyes is required to form normal binocular perception for depth and stereopsis. Binocular neurons in the striate cortex must match their orientation tuning through the two eyes in order for the animal to perceive coherently.”

15. Why was the 180 degree flexible structure made? Is it simply to imitate the shape of the eye? Are there any advantages to making a 180 degree flexible structure?

We thank the referee for this constructive question. In this work, we first report the bioinspired striate cortex with binocularity and orientation selectivity based on the crossbar array of self-powered memristors where each cross-point contains one CsFAPbI₃ perovskite solar cell and one CsPbBr₂I perovskite memristor. The second-order CsPbBr₂I memristor with mobile halogenic vacancy similar to Ca²⁺ dynamics in the striate cortical synapse, which allows the

emulation of rate-based plasticity. While the CsFAPbI₃ perovskite solar cell can be viewed as photosensory retinal cells to synapse with striate cortical cell for converting external optical signals into electrical signals. In our hardware implementation, we directly synapse artificial retinal photosensory cells with cortical cell for mimicking the artificial retina combined with striate cortex with 180° flexible structure since the human retina with hemispherical structure can excellently adapt the optical environment and reduce the complexity of optical system by directly compensating the aberration. For practical application, the integrated hemispherical structure may be surgically implanted into a human eye to help blind people regain their optesthesia.

This paper is not carefully written. Please find the minor suggestions and corrections below.

- 1) Some sentences in main text are just copied and pasted from the abstract. The first sentence of the abstract is grammatically wrong.**
- 2) 1) in line 75, synapse ==> synapses**
- 3) in line 87, the round bracket is missing.**
- 4) in line 320, wight ==> typo.**
- 5) The sentence in line 110~111 needs to be correct.**
- 6) In line 122 in supporting information, a resistor cannot have the unit “us”.**

We would like to thank reviewer for pointing out these errors. Note that these errors have been revised in the manuscript.

Reviewer #1 (Remarks to the Author):

The authors provided an in-depth response to my comments. I thank the authors for their detail answers. I agree on most of the answers but I still have a fundamental issue with the BCM implementation and the answer of the authors. The authors mentioned that situation where $\rho_x=0$ or $\rho_y=0$ are irrational. In biology, you can have pre_neuron activity ($R_x>0$) without triggering firing at the post_neuron ($R_y=0$). Post_neuron doesn't reach its threshold. Equivalently, input neuron activity can be 0 and output neuron activity can be >0 (output activity is initiated by other pre_neurons). BCM learning, as an extension of Hebbian learning should satisfy $\Delta G=0$ (synaptic conductance modulation) if $\rho_x=0$ or $\rho_y=0$. The definition of the BCM learning by the authors seems to me a specific case that fits their experiments, but it is not exactly an implementation of BCM. I would like to ask the authors to clearly mention what is the restricted BCM implementation that they are proposing in order to not propagate any wrong understanding of BCM to the reader.

Reviewer #2 (Remarks to the Author):

Authors well revised their manuscript according to reviewers' comments. In my opinion now it is acceptable for publication in this journal.

Reviewer #3 (Remarks to the Author):

The authors have addressed almost all of my concerns about the manuscript, which are mainly on the proposed memristor's characteristics. Even though the memristor does not seem to be a optimized device for the proposed application (it consumes large energy due to its high current and slow operation speed), the integration level of the device (monolithic integration of a solar cell and a memristor on the crossbar array) is high compared to the previous studies, and the experimental demonstration of the artificial vision system recognizing orientations in a hardware level makes a meaningful contribution for the field. However, there are a few concerns remaining about the array operation, which must be addressed by the authors.

1. In Figure 4a, the authors illustrated the array with memristor and solar cell monolithically integrated in a crossbar. However, each single solar cell (which can be considered as a photodetector) is connected to three memristors through a middle electrode, and it seems like every bottom electrode (three) are connected to a single neuron. In this structure, giving stimulus to a single solar cell will apply voltage to a three memristor, and the three memristors sharing the same middle electrode will be programmed together, which means that the given 3×3 array is same to a 1×3 array in that figure. How each solar cell on the memristor affects only a single memristor when the memristors share a same middle electrode? (If the solar cell makes some voltage, the middle electrode of the solar cell will have the V_{solar} , and the V_{solar} will be shared to the three memristors connected to the middle electrode, which makes impossible to differently program the three memristors.)

2. The authors claim that they used "programming-inhibit operation" which is usually called "half voltage scheme". When a half-voltage scheme is used, it is impossible to avoid some leakage current (Current at half voltage of the unselected cells) in a crossbar structure. According to the information from the authors, the device shows small on/off ratio (< 10), and in this case, the exact off current of the device in the array will not be same to the off current of the stand-alone device (The off current of the device must be $I_{selected} + (column\ size - 1) \times (I_{unselected\ at\ half-voltage})$). However, in the Figure 5c, it seems that the device can achieve 50 μS conductance, which is the off state resistance of the stand-alone device. Therefore, the authors should clarify that whether they considered the sneak path (or off-current of the unselected cells) in the simulation, or not, or, how they avoid the off current noise when they use half-voltage scheme. (Similar to the simulation results, the effect of the off current does not observed in the Figure 4g.)

3. The authors verified endurance by using I-V measurement with cc. The conventional methods for endurance test is using pulses. Then the authors might achieve better endurance data. If my comment is not true, please provide the detailed pulse condition in the manuscript.

4. The authors used constant (continuous) 0.1V bias for retention test. The authors should use pulse for retention and remove the read disturbance effect. If my comment is not true, please provide the detailed pulse condition in the manuscript.

5. For LTP-LTD data, the authors also used constant bias, which is not conventional methods. With constant bias, the authors cannot achieve meaningful LTP-LTD graphs. If my comment is not true, please provide the detailed pulse condition in the manuscript.

Response to Reviewers' Comments

Reviewer's Comment

Our Response

Changes Made in the Manuscript

Reviewer #1 (Remarks to the Author)

Comments to the Author

The authors provided an in-depth response to my comments. I thank the authors for their detail answers. I agree on most of the answers but I still have a fundamental issue with the BCM implementation and the answer of the authors. The authors mentioned that situation where $\rho_x=0$ or $\rho_y=0$ are irrational. In biology, you can have pre_neuron activity ($R_x>0$) without triggering firing at the post_neuron ($R_y=0$). Post_neuron doesn't reach its threshold. Equivalently, input neuron activity can be 0 and output neuron activity can be >0 (output activity is initiated by other pre_neurons). BCM learning, as an extension of Hebbian learning should satisfy $\Delta_G=0$ (synaptic conductance modulation) if $\rho_x=0$ or $\rho_y=0$. The definition of the BCM learning by the authors seems to me a specific case that fits their experiments, but it is not exactly an implementation of BCM. I would like to ask the authors to clearly mention what is the restricted BCM implementation that they are proposing in order to not propagate any wrong understanding of BCM to the reader.

Thanks for the reviewer's valuable comment and we apologize for our inaccurate statement that the $\rho_x=0$ or $\rho_y=0$ are irrational. In the classic monocular deprivation experiments, the neurons in the visual cortex tend to disconnect from neurons of the anesthetic-treatment deprived eye which means the retinal neurons are absolutely inactive and their corresponding fire frequency are equal to zero. Under this special condition, the inactivation of neurons actually protects the synapses of deprived eye from monocular depression and maintain the synaptic weight at origin state ($\Delta_G=0$). All these experimental results are consistent with the theoretical BCM learning rule. To avoid the misunderstanding of BCM to the reader, we clarified BCM implementation in the revised manuscript as follows:

“It should be pointed out that, this BCM implementation is based on the assumption that the

responsiveness of neuron connectivity is active which means pre-synapse and post-synapse need keep fires to active the retina-genicular-cortical pathway.”

Reviewer #2 (Remarks to the Author):

Authors well revised their manuscript according to reviewers' comments. In my opinion now it is acceptable for publication in this journal.

Thank you very much for spending your valuable time on reviewing the manuscript and for recommending the publication of our manuscript.

Reviewer #3 (Remarks to the Author):

The authors have addressed almost all of my concerns about the manuscript, which are mainly on the proposed memristor's characteristics. Even though the memristor does not seem to be a optimized device for the proposed application (it consumes large energy due to its high current and slow operation speed), the integration level of the device (monolithic integration of a solar cell and a memristor on the crossbar array) is high compared to the previous studies, and the experimental demonstration of the artificial vision system recognizing orientations in a hardware level makes a meaningful contribution for the field. However, there are a few concerns remaining about the array operation, which must be addressed by the authors.

1. In Figure 4a, the authors illustrated the array with memristor and solar cell monolithically integrated in a crossbar. However, each single solar cell (which can be considered as a photodetector) is connected to three memristors through a middle electrode, and it seems like every bottom electrode (three) are connected to a single neuron. In this structure, giving stimulus to a single solar cell will apply voltage to a three memristor, and the three memristors sharing the same middle electrode will be programmed together, which means that the given 3*3 array is same to a 1*3 array in that figure. How each solar cell on the memristor affects only a single memristor when

the memristors share a same middle electrode? (If the solar cell makes some voltage, the middle electrode of the solar cell will have the V_{solar} , and the V_{solar} will be shared to the three memristors connected to the middle electrode, which makes impossible to differently program the three memristors.)

We thank the reviewer for this question. As correctly pointed by the reviewer, the anode of each solar cell is structural connected to three memristors. However, the three memristors are belong to different electric circuits since their bottom electrodes are located on different word lines. This mutually independent structure ensures that only the memristor directly connected with stimulated solar cell are triggered and the two other memristors cannot be mis-operated.

2. The authors claim that they used “programming-inhibit operation” which is usually called “half voltage scheme”. When a half-voltage scheme is used, it is impossible to avoid some leakage current (Current at half voltage of the unselected cells) in a crossbar structure. According to the information from the authors, the device shows small on/off ratio (< 10), and in this case, the exact off current of the device in the array will not be same to the off current of the stand-alone device (The off current of the device must be $I_{\text{selected}} + (\text{column size} - 1) * (I_{\text{unselected at half-voltage}})$). However, in the Figure 5c, it seems that the device can achieve 50 uS conductance, which is the off state resistance of the stand-alone device. Therefore, the authors should clarify that whether they considered the sneak path (or off-current of the unselected cells) in the simulation, or not, or, how they avoid the off current noise when they use half-voltage scheme. (Similar to the simulation results, the effect of the off current does not observed in the Figure 4g.)

We thank the referee for the technical question. Indeed, we employed a readout process to determine the array states which can avoid leakage current, as shown in Fig. R1. During the readout process, the selected bit line was set to readout voltage of 0.1V and the other bit lines were kept to 0V, the current was readout from three word lines instantaneously. Based on this readout design, the leakage current can be avoided from the pattern learning results in Fig. 4g. In addition, we directly chose the array conductance as independent variable and the current (sneak path) is ignored in the simulation work. On the one hand, the BCM-based synaptic weight modification is directly described as the conductance change, while the current is only

the intermediate variable during array operation. On the other hand, thanks to the separated conductance programming and readout design, we can monitor and operate the array states conveniently and accurately without the disturber from sneak path. The detailed flow chart of simulation have provided in Fig. S20.

We included following discussion in the supplementary information:

“The array states are determined by the following readout process, the selected bit line was set to readout voltage of 0.1V and the other bit lines were kept to 0V, the current were readout from three word lines instantaneously.”

Figure R1. The readout method.

3. The authors verified endurance by using I-V measurement with cc. The conventional methods for endurance test is using pulses. Then the authors might achieve better endurance data. If my comment is not true, please provide the detailed pulse condition in the manuscript.

Thanks for the reviewer’s valuable comment. As suggested by reviewer, we implemented the pulse-based endurance test, as shown in Fig. R2. Under the alternate positive 0.5V/5ms and negative -0.5V/5ms pulses, the device shows potentiation and depression effects and the resistance states show almost no degradation in the 1000 cycles.

We included following discussion in the supplementary information:

“We implemented the pulse-based endurance test, as shown in Fig. S3. Under the alternate positive 0.5V/5ms and negative -0.5V/5ms pulses, the device shows potentiation and depression effects and the resistance states show almost no degradation in the 1000 cycles.”

Figure R2. The endurance characteristic of the memristor in the pulse-based test.

4. The authors used constant (continuous) 0.1V bias for retention test. The authors should use pulse for retention and remove the read disturbance effect. If my comment is not true, please provide the detailed pulse condition in the manuscript.

Thanks for the reviewer's valuable comment. We used the pulse of 0.1V/5ms for retention test and the pulse interval was set to 20s. In order to eliminate this misunderstanding, we have revised the explanation in the supplementary information as follows:

“The test pulse was set to 0.1V/5ms and the pulse interval was set to 20s.”

5. For LTP-LTD data, the authors also used constant bias, which is not conventional methods. With constant bias, the authors cannot achieve meaningful LTP-LTD graphs. If my comment is not true, please provide the detailed pulse condition in the manuscript.

Thanks for the reviewer's valuable comment. Indeed, we used the pulse of 0.5V/5ms and -0.5V/5ms for LTP and LTD test, respectively. In order to eliminate this misunderstanding, we have revised the explanation in the revised manuscript as follows:

“Fig. 2d shows that the memristor conductance can be gradually increased (decreased) by consecutive 100 positive 0.5V/5ms (negative -0.5V/5ms) pulses, which is corresponding to the synaptic LTP and LTD behaviors.”

Reviewer #1 (Remarks to the Author):

The author answered all my comments. I don't have any further question.

Reviewer #3 (Remarks to the Author):

The authors have largely addressed the concerns except the question 1. I still don't understand how to operate array with the scheme. Except that, I suggest accepting this manuscript.

Response to Reviewers' Comments

Reviewer's Comment

Our Response

Reviewer #1 (Remarks to the Author):

The authors answered all my comments. I don't have any further question.

Thank you very much for spending your valuable time on reviewing the manuscript and for recommending the publication of our manuscript.

Reviewer #3 (Remarks to the Author):

The authors have largely addressed the concerns except the question 1. I still don't understand how to operate array with the scheme. Except that, I suggest accepting this manuscript.

We thank the reviewer for this question. For the question 1, only the top electrode of three memristors are electrically connected, however, the bottom electrodes are separated which are connected to the corresponding solar cells. When a single solar cell works, the photovoltage is applied to its corresponding memristor since the cathode of solar cell and bottom electrode are connected through external circuit. The circuit loop cannot be formed with other two memristors since their bottom electrodes do not connect to the cathode of working solar cell. In addition, the other two solar cells contain large internal resistance in dark condition, which will protect its corresponding memristors from mis-operating by the adjacent circuit voltage.